# Black-Box Adversarial Attack on Dialogue Generation via Multi-Objective Optimization

## Abstract

Transformer-based dialogue generation (DG) models are ubiquitous in modern conversational artificial intelligence (AI) platforms. These models, however, are susceptible to adversarial attacks, i.e., prompts that appear textually indiscernible from normal inputs but are maliciously crafted to make the models generate responses incoherent and irrelevant to the conversational context. Evaluating the adversarial robustness of DG models is thus crucial to their real-world deployment. Adversarial methods typically exploit gradient information and output logits (or probabilities) to effectively modify key input tokens, thereby achieving excellent attack performance. Nevertheless, such white-box approaches are impractical in real-world scenarios since the models' internal parameters are typically inaccessible. While black-box methods, which exploit only input prompts and DG models' output responses to craft adversarial attacks, offer a wider applicability, they often suffer from poor performance.

In a human-machine conversation, good generated responses are expected to be semantically coherent and textually succinct. We thus formulate adversarial attack on DG models as a bi-objective optimization problem, where input prompts are modified in order to 1) minimize the response coherence, and 2) maximize the generation length. In this paper, we empirically demonstrate that optimizing either objective alone results in subpar performance. We then propose a dialogue generation attack framework (DGAttack) that employs multi-objective optimization to consider both objectives simultaneously when perturbing user prompts to craft adversarial inputs. Leveraging the exploration capability of multi-objective evolutionary algorithm due to its intrinsic diversity preservation, DGAttack successfully creates effective adversarial prompts in a true black-box manner, i.e., accessing solely DG models' inputs and outputs. Experiments across four benchmark datasets and three language models (i.e., BART, DialoGPT, T5) demonstrate the excellent performance of DGAttack compared to existing white-box, gray-box, and black-box approaches. Especially, benchmarks with large language models (i.e., Llama 3.1 and Gemma 2) suggest that DGAttack is the state-of-the-art black-box adversarial attack on dialogue generation.

## 1 Introduction

Dialogue generation (DG) has made advancing strides with pre-trained transformers (Zhang et al., 2020c; Roller et al., 2021), enabling the creation of sophisticated chatbots capable of natural, coherent conversations. Nevertheless, DG models remain vulnerable to adversarial attacks—malicious inputs that, while appearing benign, are designed to disrupt the model's output by generating incoherent or irrelevant responses (Goodfellow et al., 2015). Given the increasing deployment of DG models in real-world applications, evaluating their adversarial robustness is critical to ensuring their reliability and trustworthiness.

White-box adversarial attacks, where attackers exploit gradient information and output logits to craft adversarial inputs, have shown excellent performance in degrading the response quality (Li et al., 2023a; Cheng et al., 2018). These attacks effectively identify critical tokens and modify them to compromise the model's performance. However, white-box methods rely on access to DG models' internal parameters—information that is often unavailable in real-world settings due to proprietary

restrictions or security constraints. In contrast, black-box attacks, which do not require access to model parameters or gradients, offer broader applicability. These methods craft adversarial samples based solely on input prompts and output responses. Black-box attacks tend to underperform compared to their white-box counterparts, as they do not make use of internal knowledge.

A key challenge in attacking DG models lies in the conversational nature. Unlike other tasks where inputs are processed independently, DG models generate responses based on both the current input and the accumulated chat history (Liu et al., 2020). This reliance on prior context makes small input perturbations less effective, particularly in black-box settings. Traditional black-box adversarial methods, which typically focus on minimizing accuracy alone (Garg & Ramakrishnan, 2020; Ren et al., 2019a; Li et al., 2020; Zhang et al., 2021), struggle to fully exploit the vulnerabilities of DG models in these scenarios. Responses from conversational AI agents are expected to be relevant, coherent, and succinct. We observed that adversarial prompts that induce DG models to generate longer responses tend to have a greater attack success rate, as these extended outputs are often irrelevant to the intended conversational context. However, exploiting this trade-off is non-trivial since modern large language models (LLMs) are adept at generating coherent long-form responses. It is thus necessary to simultaneously optimize both generation length and response coherence.

To address these challenges, we propose **DGAttack**, a novel black-box adversarial attack framework that formulates attacking DG models as a bi-objective optimization problem. Employing the non-dominated sorting genetic algorithm II (NSGA-II) (Deb et al., 2002), DGAttack simultaneously optimizes two objectives: maximizing generation length and minimizing response coherence. This approach allows us to explore the adversarial space efficiently while relying solely on the model's inputs and outputs—making it particularly suitable for black-box settings, where internal parameters and output probabilities are inaccessible. Through comprehensive experiments on four benchmark datasets and three language models (BART, DialoGPT, and T5), we demonstrate that DGAttack outperforms existing black-box, white-box, and gray-box adversarial methods. Moreover, DGAttack sets a new standard for black-box adversarial attacks on dialogue generation, particularly when applied to large language models like Llama 3.1 and Gemma 2, demonstrating state-of-the-art performance in generating effective adversarial prompts.

## 2 RELATED WORKS

### 2.1 DIALOGUE GENERATION

Dialogue generation (DG) involves the task of processing natural language inputs and producing human-like responses, typically in the context of ongoing conversations, such as interactions with chatbots. Typically, a DG model must interpret the conversation history up to the current turn and generate appropriate responses in a structured manner. Over the past few years, DG has seen significant progress, particularly with pre-trained transformer-based models, such as decoder-only models like DialoGPT (Zhang et al., 2020b) and Llama (Touvron et al., 2023), as well as encoder-decoder models like T5 (Raffel et al., 2020) and BART (Lewis et al., 2020). These models generate responses that resemble natural human dialogue, with some even utilizing additional information, such as user profiles or conversational context, to create more personalized and context-aware interactions.

### 2.2 TEXTUAL ADVERSARIAL ATTACKS

Textual adversarial attacks can be used for testing the robustness of natural language processing models. These attacks are categorized into character-level, word-level, and sentence-level approaches (Papernot et al., 2016; Ebrahimi et al., 2018; Li et al., 2018; Chen et al., 2022). Early character-level attacks manipulated individual characters—by adding, deleting, or substituting them—which allowed for straightforward adversarial sample generation (Belinkov & Bisk, 2018). However, these approaches often resulted in grammatically incorrect outputs, making them susceptible to grammar-based defense mechanisms (Pruthi et al., 2019). Consequently, character-level attacks have become less prominent in recent works (Le et al., 2022). Sentence-level attacks, which perturb entire sentences, offer better grammatical correctness by employing techniques such as paraphrasing and encoding-decoding (Iyyer et al., 2018; Zhao et al., 2017). Despite their syntactic accuracy, these methods often introduce substantial semantic shifts, reducing the overall success rate of the attack. Word-level attacks have emerged as a popular approach due to their ability to balance

grammatical accuracy, semantic coherence, and attack success. These methods typically involve word substitution, addition, or deletion while preserving the overall meaning and context of the sentence (Jin et al., 2019; Ren et al., 2019b). Such strategies offer a middle ground between maintaining meaning and generating effective adversarial samples.

Recent advancements in learning-based methods, particularly using BERT-based Masked Language Models (MLMs), have improved the semantic relevance of adversarial samples by leveraging context to generate word substitutions (Garg & Ramakrishnan, 2020; Li et al., 2020). However, these models can still introduce ambiguity in tasks like rumor detection and sentiment analysis.

While most adversarial attacks focus on classification, there is a growing interest in sequence-to-sequence (seq2seq) models. Works like NMTSloth (Chen et al., 2022) target length manipulation in neural machine translation (NMT) systems, aiming to generate longer and less coherent translations. Seq2Sick (Cheng et al., 2018) and other methods attempt to degrade the generation confidence in seq2seq tasks by reducing the likelihood of producing correct outputs (Michel et al., 2019).

Most notably, multi-objective white-box attacks have been applied to dialogue generation models, where approaches such as DGSlow (Li et al., 2023b) optimize for both accuracy minimization and generation length maximization. While white-box attacks utilize gradient information or model parameters to effectively create adversarial prompts, such assumptions on the accessibility of internal knowledge do not hold in practice. Several gradient-free attacks on NLP models make use of the models' output logits or probabilities for importance ranking in order to identify key input tokens for perturbation (Garg & Ramakrishnan, 2020; Li et al., 2020; Ren et al., 2019a). However, neither logit nor probability information is available during interactions with real-world conversational agents.

## 3 METHODOLOGY

### 3.1 PROBLEM STATEMENT

#### 3.1.1 DIALOGUE GENERATION

Suppose a chatbot aims to model conversations between two individuals. We follow a similar setup (Liu et al., 2020), where each individual has a persona (e.g., $c_A$ for person A), described with $L$ profile sentences $c_{A_1}, \ldots, c_{A_L}$. Person A chats with another person B through an $N$-turn dialogue $(x_{A_1}, x_{B_1}, \ldots, x_{A_N}, x_{B_N})$, where $N$ is the total number of turns and $x_{A_n}$ is the utterance that A says in the $n$-th turn. A DG model $f$ takes the persona $c_A$, the entire dialogue history until the $n$-th turn $h_{A_n} = (x_{B_1}, \ldots, x_{A_{n-1}})$, and B's current utterance $x_{B_n}$ as inputs, generating outputs $x_{A_n}$ by maximizing the probability $p(x_{A_n} | c_A, h_{A_n}, x_{B_n})$. The same process applies for B to keep the conversation going.

#### 3.1.2 DIALOGUE GENERATION ADVERSARIAL ATTACK

In each dialogue turn $n$, we craft an adversarial utterance $x_{B_n}$ for person B, with the goal of misleading the chatbot designed to emulate person A. It is crucial to maintain the integrity of the chat history $h_{A_n} = (x_{B_1}, \ldots, x_{A_{n-1}})$, ensuring that it remains unchanged to reflect realistic conditions in practical applications.

An optimal DG adversarial sample in the $n$-th turn is an utterance $x_{B_n}^*$:

$$x_{B_n}^* = \arg\min_{\hat{x}_{B_n}} M(x_{\mathrm{ref}_n}, \hat{x}_{A_n})$$

$$\text{subject to:} \quad \hat{x}_{A_n} = f(c_A, h_{A_n}, \hat{x}_{B_n}) \quad \text{and} \quad \rho(x_{B_n}, \hat{x}_{B_n}) > \epsilon \tag{1}$$

$$\hat{x}_{B_n} = x_{B_n} + \Delta x_{B_n}$$

where $\rho(.)$ is a similarity function and $\epsilon$ is the similarity threshold between the original input $x_{B_n}$ and the crafted adversarial utterance $\hat{x}_{B_n}$. Here, $\Delta x_{B_n}$ represents the perturbation applied to the original utterance. $M(\cdot)$ is typically measured using neural machine translation (NMT) metrics, such as BLEU (Papineni et al., 2002), METEOR (Banerjee & Lavie, 2005), and ROUGE (Lin & Och, 2004), to evaluate the quality of the output response $\hat{x}_{A_n}$ relative to a reference response $x_{\mathrm{ref}_n}$.

In dialogue generation, longer generated responses are often observed to drift away from the original context and introduce irrelevant or nonsensical content, making them particularly effective for ad-

versarial attacks. However, achieving longer responses presents a challenge, as language models are trained to maintain coherence and relevance, even when generating lengthy sequences. To address this, we define two primary objectives for our black-box adversarial attack: **Accuracy Score (AS)** and **Generation Length (GL)**.

**Accuracy Score** represents the degradation in the model's accuracy. It is calculated as the combined sum of accuracy metrics—BLEU, ROUGE, and METEOR—by comparing the adversarially generated response $\hat{x}_{A_n}$ to the original, unperturbed response $x_{A_n}$. This objective measures the reduction in similarity between the original and adversarially generated responses:

$$\text{AS}(\hat{x}_{B_n}) = \text{BLEU}(\hat{x}_{A_n}, x_{A_n}) + \text{ROUGE}(\hat{x}_{A_n}, x_{A_n}) + \text{METEOR}(\hat{x}_{A_n}, x_{A_n}) \tag{2}$$

**Generation Length** is introduced as the second objective. Since generating longer outputs can lead to semantically less accurate responses, GL is defined as the total number of tokens in the generated output sentence $\hat{x}_{A_n}$, representing the length of the adversarial response:

$$\text{GL}(\hat{x}_{B_n}) = |\hat{x}_{A_n}| \tag{3}$$

These two objectives are optimized simultaneously to craft adversarial samples that force DG models to generate responses that are not only inaccurate but also longer and more irrelevant.

In white-box adversarial attack (Li et al., 2023b), the accuracy objective was defined using cumulative probabilities with respect to a reference response $x_{\text{ref}_n}$, known as **Targeted Confidence (TC)**:

$$\text{TC}(\hat{x}_{B_n}) = \sum_{t=1}^{|x_{\text{ref}_n}|} p(x_{\text{ref}_n,t}|c_A, h_{A_n}, \hat{x}_{B_n}, x_{\text{ref}_n,<t}) \tag{4}$$

Minimizing TC reduces the likelihood of the model generating the reference response $x_{\text{ref}_n}$. However, in real-world scenarios, accessing internal model probabilities and reference responses is typically infeasible. The only available feedback is the generated output from the target DG model. To overcome this limitation, we redefine the accuracy objective for our black-box setting by leveraging the model's original generated response from the unperturbed input sentence as a pseudo-reference. Instead of comparing the adversarial response $\hat{x}_{A_n}$ to an external reference $x_{\text{ref}_n}$, we compare it to the original response $x_{A_n}$ generated by the model in response to the unperturbed input $x_{B_n}$. This approach allows us to practically evaluate accuracy in black-box settings by using the model's own outputs as a baseline for assessing adversarial success. The goal is to minimize the similarity between the adversarial response $\hat{x}_{A_n}$ and the original response $x_{A_n}$, thereby degrading the model's performance while ensuring the adversarial input remains contextually appropriate.

## 3.2 ADVERSARIAL ATTACK VIA MULTI-OBJECTIVE OPTIMIZATION

### 3.2.1 PARETO DOMINANCE IN ADVERSARIAL ATTACK ON DIALOGUE GENERATION

Regarding the two objectives, Accuracy Score (AS) and Generation Length (GL), a candidate adversarial sentence $x_a$ is said to Pareto dominate another adversarial sentence $x_b$ (denoted as $x_a \succ x_b$) if $x_a$ is no worse than $x_b$ in both objectives and strictly better in at least one objective:

$$x_a \succ x_b \Leftrightarrow (\text{AS}(x_a) \leq \text{AS}(x_b) \wedge \text{GL}(x_a) \geq \text{GL}(x_b)) \wedge (\text{AS}(x_a) < \text{AS}(x_b) \vee \text{GL}(x_a) > \text{GL}(x_b)) \tag{5}$$

The *utopian* adversarial sentence that force DG models to generate responses with the maximal length and the minimal accuracy is hard to obtain. This is because maximizing GL does not always succeed in minimizing accuracy, especially regarding modern LLMs, and while minimizing accuracy could unintentionally shorten responses. Instead, multi-objective attack aims to obtain the Pareto set of adversarial sentences that are all optimal in the sense that they are not Pareto dominated by any sentences in the adversarial space. The Pareto set forms a Pareto front in the objective space (GL,AS), as illustrated in Fig. 1, where each Pareto-optimal sentence represent an optimal trade-off

between response length and accuracy. In practice, we do not need to obtain the entire Pareto set but just an approximation set of Pareto-optimal sentences that are well spread on the Pareto front.

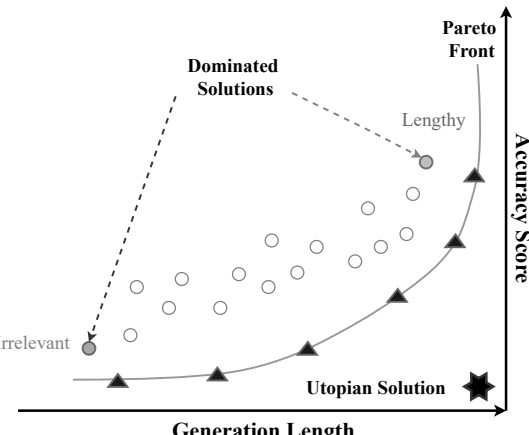

Figure 1: Illustration of the Pareto front in the objective space regarding the two objectives: maximizing generation length (GL) and minimizing accuracy score (AS). Candidate sentences on the Pareto front are not Pareto dominated by any feasible sentences. Instead of searching for the entire Pareto set, we aim to obtain an approximation set of non-dominated sentences that together approximate well the Pareto front (depicted as black triangle).

Pareto dominance-based optimization allows us to straightforwardly optimize the two objectives at the same time but separately, rather than aggregating them into a single objective as in (Li et al., 2023b). Hyperparameter tuning for the aggregation weights of AS and GL is non-trivial because the proper weights depend on the ranges of the objectives and the accuracy metrics, as well as the specific conversation under attack. Evolutionary algorithms, due to their population-based operation, are well-suited to directly searching for an approximation set of diverse adversarial sentences.

### 3.2.2 CRAFTING ADVERSARIAL ATTACKS WITH MULTI-OBJECTIVE GENETIC ALGORITHM

In our DGAttack framework, we adopt the non-dominated sorting genetic algorithm II (NSGA-II) (Deb et al., 2002) to obtain a good approximation set of diverse non-dominated adversarial sentences. Figure 2 illustrates the workflow of DGAttack.

In the first generation $t = 1$, DGAttack generates the initial population $P^1$ consisting of $N$ adversarial sentences, which are created from random perturbations of the original input sentence $x_{B_n}$. In each generation $t$, promising candidate sentences from $P^t$ (in terms of Pareto dominance regarding the two objectives AS and GL) are copied into a selection set $S^t$. Two variation operators (crossover and mutation) are applied on $S^t$ to create a set $O^t$ of new candidates. Crossover recombines each pair of selected sentences $x, x' \in S^t$ (i.e., parents) by exchanging random segments of their words to craft two new sentences $o, o' \in O^t$ (i.e., offspring). Mutation randomly perturbs some words in offspring sentences $o \in O^t$, thereby introducing novel tokens to the search process. After variation, the current population and the offspring population are merged into a pool $(P^t + O^t)$ from which the non-dominated sorting procedure assigns a rank to each candidate based on their Pareto dominance. Another selection round is conducted to select candidates into the next generation $P^{t+1}$ based on their ranks. If candidates from the same rank compete, the ones with higher crowding distances is preferred (i.e., the ones that are far from others). The above procedure of *selection - variation* is iterated until the allowed number of generations is reached. In this final population, the candidates that are not dominated by others are regarded as the approximation set of non-dominated adversarial sentences obtained by DGAttack. Further details can be found in Appendix B.

**Perturbation Strategy.** This strategy is applied in both initialization and mutation steps to introduce adversarial perturbations. To generate new sentences, we perturb salient words within existing sentences. Salient words are identified using POS tags, focusing on nouns, adjectives, and verbs,

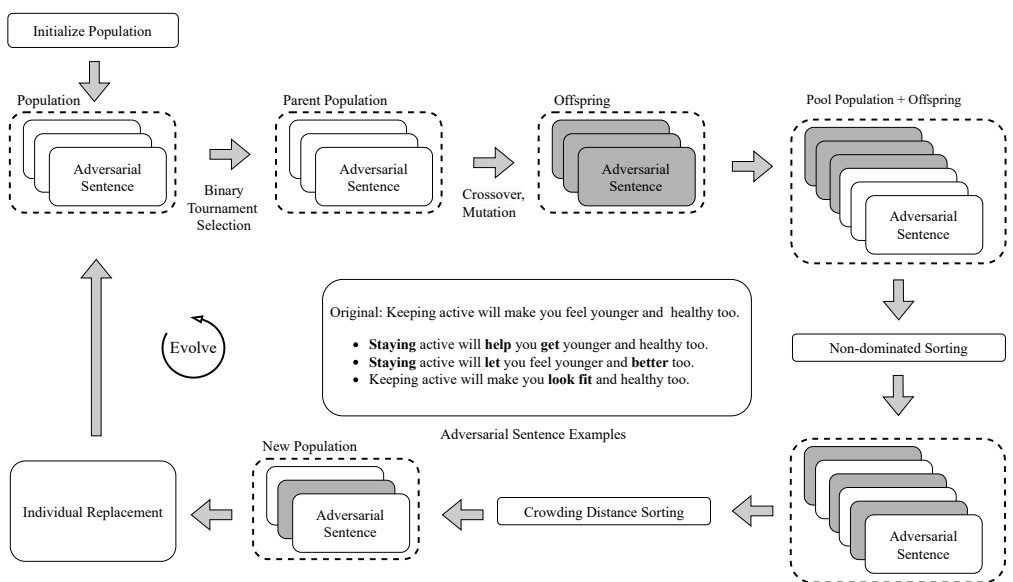

Figure 2: The main framework of DGAttack and advesarial sentence examples.

i.e., parts of speech that affect the sentences' meanings the most. To maintain sentence coherence, we exclude immutable stopwords (e.g., auxiliary verbs and common pronouns) from perturbation.

Our perturbation strategy generates adversarial samples that are more fluent than traditional rule-based substitutions. We use a pre-trained BERT model (Devlin et al., 2019) to predict contextually appropriate replacements for salient words. The process starts by replacing salient words with a [MASK] token. For example, a sentence with the word $w_i$ masked would be transformed into $s_{wi} = [w_0, \ldots, w_{i-1}, [MASK], w_{i+1}, \ldots]$. We then craft adversarial sentences by filling the [MASK] token with BERT's predictions. BERT-MLM is a powerful pre-trained language model, and its predicted tokens generally fit well into the grammar and context of the text.

However, BERT does not guarantee semantic coherence, as an alternative word can fit both grammatically and contextually while still having a different meaning. To address this issue, we filter out low semantic similarity candidates using the Universal Sentence Encoder (USE) (Cer et al., 2018) sentence similarity function, retaining only candidates with high similarity to the original sentence. We also check those BERT predictions by filtering out antonyms using the WordNet (Miller, 1995), enhancing similarity between adversarial samples and original sentences. We also use Language-Tool[1], an open-source grammar checker to filter out potential grammatical errors.

## 4 EXPERIMENTS

### 4.1 EXPERIMENTS SETTINGS

#### 4.1.1 DATASETS

Our experimental setup closely follows the methodology described in the white-box approach DGSlow Li et al. (2023b). We evaluate our method on four benchmark datasets: Blended Skill Talk (BST) (Smith et al., 2020), Persona Chat (PC) (Zhang et al., 2018), ConvAI2 (CV2) (Dinan et al., 2019), and Empathetic Dialogues (ED) (Rashkin et al., 2019). These datasets are preprocessed for dialogue generation (DG) tasks following settings outlined in Section 3.1.1. The statistics of the datasets (training sets) are shown in Appendix A.1.

---

[1]https://languagetool.org/

### 4.1.2 TARGET MODELS

We target three DG models: DialoGPT (Zhang et al., 2020b), BART (Lewis et al., 2020), and T5 (Raffel et al., 2020). DialoGPT is a pretrained transformer based on GPT-2 (Radford et al., 2018), specifically trained on Reddit comments for dialogue response generation. BART and T5 are seq2seq encoder-decoder models pretrained on diverse and open-domain datasets. Details of the performance of Target Models are shown in Appendix A.2.

Following established practices in the field, we utilize Byte-level BPE tokenization (Radford et al., 2019) pre-trained on open-domain datasets, as implemented in HuggingFace tokenizers. To meet the DG requirements, we incorporate two additional special tokens, namely, [PS] and [SEP]. The [PS] token is inserted before each persona to help the model recognize the personality of each speaker. The [SEP] token is used to separate utterances within a dialogue, allowing the model to understand the structural information within the chat history.

### 4.1.3 METRICS

We evaluate our method based on generation length, accuracy metrics, and Attack Success Rate (ASR) of the generated responses to adversarial samples as in (Li et al., 2023b). We employ three standard NLP accuracy metrics BLEU (Papineni et al., 2002), ROUGE (Lin & Och, 2004), and METEOR (Banerjee & Lavie, 2005). These metrics quantify the correspondence between the generated responses and the reference outputs. The ASR metric is defined as in (Li et al., 2023b):

$$\text{ASR} = \frac{1}{N} \sum_{i=1}^{N} 1[\cos(x_i, \hat{x}_i) > \epsilon \wedge E(y_i, \hat{y}_i) > \tau], \tag{6}$$
$$\text{subject to:} \quad E(y, \hat{y}) = M(y, y_{\text{ref}}) - M(\hat{y}, y_{\text{ref}}).$$

where $\cos(.)$ denotes the cosine similarity between the embeddings of the original input $x$ and the crafted input $\hat{x}$. $M(\cdot, \cdot)$ represents the average score of the three accuracy metrics. An attack is considered successful if the adversarial input induces a more irrelevant output ($> \tau$) while preserving sufficient semantics of the original input ($> \epsilon$). Details of the hyperparameters can be found in Appendix A.3

### 4.1.4 BASELINES

We evaluate our approach against two white-box and two black-box adversarial attack strategies, adapted to the dialogue generation task.

For the white-box attacks, we focus on: **1) HotFlip** (Ebrahimi et al., 2018), which generates adversarial examples through both word and character-level substitutions driven by embedding gradients. **2) TextBugger** (Li et al., 2018), which employs a greedy strategy for word substitution and character manipulation to execute white-box adversarial attacks.

We compare our methods with two recent black-box textual word-level adversarial attacks: **1) BAE (BERT-based Adversarial Examples)** (Garg & Ramakrishnan, 2020), which estimates the importance of each token by computing the change in output probability before and after deleting that token, and then uses BERT to perturb the most vulnerable words. **2) PWWS (Probability Weighted Word Saliency)** (Ren et al., 2019a), which generates adversarial examples by replacing key words with their synonyms, selected according to a probability-weighted saliency score, aiming to mislead the model while preserving the original meaning of the sentence. Although both of these methods are labeled as black-box approaches, we argue that they belong to the gray-box category because they rely on access to the model's output probabilities to identify vulnerable words. In real-world scenarios, such output probabilities are often inaccessible, limiting their practical applicability. This distinction is crucial, as truly black-box methods, like DGAttack, operate solely on the generated responses without requiring internal model details. To adapt these methods to the DG setting, we calculate the importance score based on the Targeted Confidence which is formulated as the cumulative probabilities of a sequence with respect to its reference sentence $x_{\text{ref}_n}$.

We also compare our approach with DGSlow (Li et al., 2023b), which is the state-of-the-art multi-objective white-box adversarial attack for dialogue generation. We adapt DGSlow's objectives to an

**DGAttack gray-box** baseline, in which we experiment DGAttack with TC as the accuracy objective and GL as the length objective. This adaptation underscores the effectiveness of our approach in degrading the target model's performance without requiring access to the model's parameters nor its output probabilities. Additionally, for black-box baselines, we implement a **single-objective Genetic Algorithm (GA)** targeting either accuracy score or generation length (see Appendix C).

## 4.2 EXPERIMENTAL RESULTS

Table 1: Evaluation of white-box, gray-box , and black-box attack methods on three target models across four datasets. GL denotes the average generation output length. Cos. stands for the cosine similarity between original and adversarial samples. ROU. (%) and MET. (%) denote ROUGE-L and METEOR respectively. **Bold** numbers mean the best metric values across methods.

| Dataset | Method | DialoGPT | | | | | | Bart | | | | | | T5 | | | | | |
|---|---|---|---|---|---|---|---|---|---|---|---|---|---|---|---|---|---|---|---|
| | | GL↑ | BLEU↓ | ROU.↓ | MET.↓ | ASR↑ | Cos.↑ | GL↑ | BLEU↓ | ROU.↓ | MET.↓ | ASR↑ | Cos.↑ | GL↑ | BLEU↓ | ROU.↓ | MET.↓ | ASR↑ | Cos.↑ |
| BST | FD | 16.70 | 13.74 | 18.31 | 24.00 | 39.29 | 0.79 | 16.60 | 12.74 | 18.62 | 19.41 | 25.14 | 0.88 | 14.74 | 13.30 | 21.42 | 21.03 | 17.14 | 0.90 |
| | HotFlip | 16.13 | 14.12 | 19.24 | 22.74 | 30.36 | 0.81 | 16.86 | 12.82 | 18.70 | 19.73 | 22.86 | 0.89 | 14.90 | 13.01 | 20.74 | 20.42 | 19.43 | 0.90 |
| | DGSlow | 25.54 | 9.14 | 17.03 | 22.61 | 71.43 | 0.90 | 23.50 | 8.39 | 16.37 | 19.40 | 48.00 | 0.92 | 28.69 | 9.11 | 15.82 | 19.21 | 57.14 | 0.93 |
| | PWWS | 15.30 | 13.47 | 20.10 | 25.77 | 27.61 | 0.75 | 19.86 | 11.23 | 20.27 | 23.57 | 21.61 | 0.78 | 14.12 | 13.80 | 21.67 | 20.77 | 43.86 | 0.77 |
| | BAE | 16.44 | 14.70 | 22.50 | 25.33 | 30.35 | 0.77 | 19.59 | 12.00 | 21.00 | 23.47 | 49.43 | 0.93 | 15.45 | 13.20 | 21.10 | 20.83 | 59.26 | 0.78 |
| | GA (AS) | 16.92 | 13.37 | 20.07 | 23.07 | 45.03 | 0.83 | 17.38 | 12.50 | 21.37 | 22.97 | 64.43 | 0.86 | 12.91 | 15.43 | 23.20 | 21.77 | 34.94 | 0.88 |
| | GA (GL) | 19.53 | 14.30 | 19.27 | 22.73 | 50.47 | 0.84 | 28.27 | 8.43 | 18.93 | 24.80 | 58.45 | 0.85 | 19.21 | 10.90 | 20.30 | 21.53 | 44.76 | 0.82 |
| | DGAttack | 21.76 | 13.10 | 19.47 | 22.37 | 48.13 | 0.82 | 27.64 | 8.63 | 18.07 | 23.70 | 70.03 | 0.82 | 16.71 | 12.47 | 20.63 | 20.40 | 51.00 | 0.82 |
| | **DGAttack** | 22.00 | 13.20 | 19.10 | 22.37 | 52.45 | 0.81 | 28.26 | 8.03 | 18.30 | 22.97 | 70.83 | 0.81 | 19.71 | 10.30 | 18.97 | 20.20 | 69.05 | 0.83 |
| CV2 | FD | 15.74 | 12.54 | 14.33 | 8.13 | 38.10 | -.78 | 12.30 | 10.81 | 10.52 | 11.14 | 20.13 | 0.88 | 13.97 | 9.91 | 10.62 | 9.53 | 16.78 | 0.90 |
| | HotFlip | 16.38 | 13.33 | 15.21 | 9.42 | 33.33 | 0.81 | 13.46 | 10.50 | 10.41 | 11.71 | 32.89 | 0.86 | 14.03 | 9.63 | 10.12 | 9.50 | 26.17 | 0.86 |
| | DGSlow | 28.54 | 11.70 | 13.71 | 8.00 | 64.29 | 0.81 | 23.84 | 6.51 | 8.34 | 10.52 | 56.61 | 0.87 | 28.43 | 7.74 | 8.43 | 7.71 | 53.02 | 0.88 |
| | BAE | 16.74 | 13.38 | 16.16 | 10.17 | 42.24 | 0.84 | 12.79 | 12.20 | 10.80 | 11.53 | 21.33 | 0.92 | 12.73 | 11.03 | 10.37 | 10.73 | 32.38 | 0.79 |
| | PWWS | 18.61 | 13.27 | 14.47 | 14.07 | 24.74 | 0.73 | 13.78 | 10.40 | 10.67 | 12.73 | 22.99 | 0.77 | 11.25 | 12.10 | 11.57 | 10.33 | 36.81 | 0.79 |
| | GA (AS) | 14.07 | 13.57 | 15.30 | 10.37 | 35.70 | 0.82 | 10.82 | 13.30 | 11.37 | 11.47 | 31.67 | 0.88 | 12.98 | 13.27 | 10.53 | 10.40 | 38.27 | 0.84 |
| | GA (GL) | 21.95 | 12.33 | 17.03 | 10.58 | 32.65 | 0.83 | 18.64 | 8.40 | 9.67 | 11.70 | 61.27 | 0.82 | 13.32 | 10.53 | 11.10 | 10.73 | 48.62 | 0.82 |
| | DGAttack | 23.03 | 12.80 | 15.95 | 9.99 | 34.57 | 0.82 | 19.75 | 8.23 | 9.37 | 11.40 | 50.03 | 0.82 | 13.32 | 10.83 | 11.07 | 10.43 | 30.98 | 0.81 |
| | **DGAttack** | 23.94 | 12.53 | 16.43 | 9.73 | 43.74 | 0.80 | 19.78 | 7.93 | 9.93 | 10.93 | 52.99 | 0.81 | 15.57 | 9.93 | 10.27 | 9.80 | 41.22 | 0.82 |
| PC | FD | 17.27 | 17.13 | 30.22 | 29.21 | 36.67 | 0.79 | 17.20 | 15.71 | 26.90 | 30.32 | 46.55 | 0.79 | 14.54 | 16.34 | 27.69 | 28.03 | 33.62 | 0.82 |
| | HotFlip | 17.22 | 17.74 | 28.81 | 27.92 | 56.67 | 0.79 | 17.51 | 15.01 | 26.53 | 30.34 | 57.76 | 0.77 | 14.03 | 16.13 | 27.20 | 28.37 | 43.10 | 0.81 |
| | DGSlow | 25.72 | 15.68 | 27.77 | 28.50 | 70.00 | 0.86 | 31.94 | 9.32 | 20.50 | 29.76 | 96.55 | 0.89 | 32.17 | 8.86 | 15.38 | 25.60 | 90.33 | 0.86 |
| | BAE | 16.50 | 18.93 | 29.27 | 32.07 | 52.50 | 0.79 | 16.22 | 16.17 | 27.20 | 30.80 | 39.58 | 0.92 | 14.95 | 16.13 | 27.47 | 29.83 | 44.15 | 0.82 |
| | PWWS | 16.48 | 17.67 | 30.63 | 31.70 | 39.94 | 0.71 | 17.34 | 15.90 | 26.27 | 33.37 | 50.43 | 0.80 | 13.46 | 15.77 | 28.53 | 28.20 | 32.23 | 0.79 |
| | GA (AS) | 12.38 | 20.33 | 30.47 | 30.47 | 47.66 | 0.84 | 14.11 | 17.87 | 28.33 | 29.67 | 55.39 | 0.85 | 11.95 | 17.87 | 29.83 | 28.43 | 48.92 | 0.82 |
| | GA (GL) | 18.45 | 17.91 | 28.73 | 29.37 | 48.31 | 0.81 | 25.58 | 10.80 | 23.33 | 31.47 | 73.52 | 0.81 | 18.23 | 12.80 | 26.87 | 28.57 | 62.66 | 0.81 |
| | DGAttack | 19.59 | 17.90 | 28.00 | 29.23 | 42.85 | 0.82 | 25.11 | 10.57 | 23.43 | 30.36 | 64.82 | 0.81 | 14.93 | 14.97 | 26.30 | 28.33 | 43.46 | 0.81 |
| | **DGAttack** | 19.62 | 17.43 | 28.33 | 28.93 | 48.16 | 0.79 | 25.77 | 10.13 | 22.87 | 30.67 | 66.86 | 0.82 | 18.31 | 12.37 | 26.13 | 28.87 | 50.87 | 0.80 |
| ED | FD | 15.00 | 9.03 | 12.62 | 11.06 | 41.82 | 0.75 | 19.66 | 6.54 | 10.44 | 11.03 | 44.26 | 0.76 | 16.66 | 7.41 | 11.30 | 11.04 | 32.79 | 0.79 |
| | HotFlip | 17.69 | 8.71 | 12.92 | 9.82 | 40.74 | 0.78 | 21.38 | 6.71 | 10.74 | 13.42 | 67.21 | 0.70 | 17.30 | 7.03 | 10.81 | 10.53 | 37.70 | 0.80 |
| | DGSlow | 24.72 | 8.93 | 12.12 | 9.66 | 69.81 | 0.90 | 34.28 | 4.22 | 8.11 | 9.70 | 98.36 | 0.82 | 38.82 | 4.02 | 6.10 | 9.91 | 94.16 | 0.92 |
| | BAE | 16.15 | 9.27 | 15.50 | 13.50 | 56.40 | 0.83 | 26.95 | 8.47 | 10.63 | 13.33 | 69.51 | 0.82 | 14.45 | 7.70 | 11.67 | 12.43 | 41.62 | 0.83 |
| | PWWS | 17.58 | 9.63 | 14.15 | 14.87 | 42.24 | 0.72 | 19.39 | 9.10 | 11.73 | 14.17 | 51.98 | 0.78 | 12.99 | 8.17 | 11.57 | 12.72 | 24.53 | 0.77 |
| | GA (AS) | 11.21 | 9.47 | 14.23 | 13.03 | 37.40 | 0.86 | 15.33 | 8.30 | 12.50 | 12.83 | 51.80 | 0.90 | 12.69 | 9.30 | 15.40 | 13.23 | 56.83 | 0.84 |
| | GA (GL) | 18.30 | 9.50 | 12.47 | 14.73 | 48.55 | 0.85 | 27.45 | 7.80 | 11.23 | 14.10 | 74.30 | 0.84 | 18.62 | 7.20 | 11.30 | 11.40 | 63.33 | 0.84 |
| | DGAttack | 19.11 | 9.42 | 12.10 | 12.17 | 42.24 | 0.82 | 26.77 | 5.43 | 9.93 | 12.83 | 68.63 | 0.82 | 18.32 | 7.47 | 10.93 | 10.53 | 46.93 | 0.81 |
| | **DGAttack** | 19.80 | 9.43 | 11.67 | 11.80 | 48.91 | 0.81 | 27.68 | 5.27 | 9.13 | 11.57 | 69.22 | 0.81 | 18.53 | 7.07 | 10.37 | 10.47 | 63.11 | 0.82 |

Our main results, shown in Table 1, outline the attack success rate, accuracy metrics, and cosine similarity. DGAttack consistently induces DG models to generate longer and less accurate responses compared to white-box, gray-box and black-box baselines. Notably, the multi-objective approach employed by DGAttack outperforms the single-objective GA in terms of overall attack effectiveness. Indeed, simultaneously targeting both response coherence and generation length leads to more powerful and disruptive adversarial attacks.

We compare the black-box DGAttack with a gray-box variant, which minimizes accuracy by leveraging the model's output probabilities. The results show that, even without access to internal information, the black-box DGAttack is capable of crafting adversarial sentences inducing DG models to generate longer and less accurate responses than the gray-box one. This can be attributed to the fact that using accuracy metrics like BLEU, ROUGE, and METEOR in DGAttack evaluates the overall coherence and fluency of the entire response, whereas the gray-box approach relies on token-level probabilities, which often capture only local confidence at the word level. It emphasizes the practicality and robustness of our method, demonstrating its effectiveness in real-world dialogue generation scenarios where access to model-specific knowledge may be restricted or unavailable.

In some cases, DGAttack performs moderately better than the white-box multi-objective method DGSlow on certain metrics. However, while DGSlow generally outperforms all other baselines and our proposed black-box methods, it cannot be used in real-world scenarios because it requires access to internal information about the target models, such as gradients or probabilities. In contrast, our black-box DGAttack does not rely on such internal parameters or even output logits, making it a feasible approach in practice where model information is unknown.

Experiment results demonstrate that DGAttack is a powerful and flexible tool for generating adversarial examples in the black-box setting. It effectively balances the dual objectives of degrading accuracy and extending generation length, producing adversarial samples that are both diverse and

impactful. DGAttack also preserves a reasonable degree of semantic coherence, as evidenced by acceptable cosine similarity scores. This combination of attack effectiveness and the maintenance of some coherence underscores the robustness of the multi-objective optimization strategy. Details on our design choices are provided in the ablation study in Appendix I.

### 4.3 QUALITATIVE RESULTS

We present a case study that demonstrates adversarial samples generated by our method. Table 2 showcases several input sentences crafted by DGAttack, which forces BART to produce lengthy and incoherent responses. By perturbing selected words in the input, our approach degrades the relevance and coherence of the generated dialogue while maintaining fluency. Notably, even with minimal token replacements—substituting words with similar meanings and part-of-speech tags—our methods induce the model to generate much longer, more irrelevant sequences $\hat{x}_{A_n}$ compared to the original responses $x_{A_n}$. These limited perturbations preserve the readability and semantic integrity of the adversarial samples, while still causing a significant quality drop in the generated responses.

Table 2: DGAttack generates adversarial sentences that force BART to produce lengthy and incoherent responses. *Italics* text represents added tokens, while ~~strikethrough~~ indicates removed tokens.

| |
| --- |
| **Persona** $c^A$: I am a grandparent at 44. |
| **Chat history** $h$: |
| **[PERSON B]** Congratulations! That's great. Do you have any hobbies? |
| **[PERSON A]** Thank you! I try to not feel old haha. Yes I love playing soccers with my friends quite often. |
| $[x_2^B \rightarrow \hat{x}_2^B]$ ~~Keeping~~ *Staying* active will make you feel ~~younger~~ *strong* and healthy too. Will you teach your ~~grandchild~~ *kids* soccer? |
| $[x_2^A]$ I think it will be something that he will really enjoy, since he's on a team at school |
| $[\hat{x}_2^A]$ I don't have a great idea, but it seems like it would be a good way to keep him active. I have a great idea, how would you feel about teaching him soccer? |
| $[x_3^B \rightarrow \hat{x}_3^B]$ ~~Wonderful~~ *Awesome*. I am ~~sure~~ *certain* they will appreciate that! I like to photograph nature in my free time. |
| $[x_3^A]$ Nature is a beautiful thing to behold. What nature is your favorite? |
| $[\hat{x}_3^A]$ I love taking pictures of everything! I have a collection of old silver/gold coins, and a few tropical fish. What is your favorite animal? |
| $[x_4^B \rightarrow \hat{x}_4^B]$ That's okay! Yes, I mostly ~~photograph nature~~ *shoot landscapes*, but I try to ~~photograph~~ *capture* other things as well, like people and ~~landmarks~~ *places*. |
| $[x_4^A]$ That's interesting. What type of things do you like to photograph? |
| $[\hat{x}_4^A]$ That's interesting. I think people and nature are the same thing. I think it's amazing how nature can see each other in so many different ways. |

### 4.4 ADVERSARIAL ATTACK AGAINST LLMS

Most adversarial attacks on LLMs, particularly in real-world deployments via API access, cannot rely on white-box techniques, which require access to gradients and model internals. Instead, existing white-box approaches often resort to transferability attacks, where an adversarial attack is first performed on an open-source LLM, and then transferred to other LLMs. In contrast, our black-box DGAttack can directly targets LLMs without requiring access to their gradients or output logits, making it well-suited for attacking LLMs deployed through APIs. This is a critical advantage, as most real-world LLMs, apart from open-source models, do not expose internal information, making gradient-based attacks impractical.

Table 3 demonstrates that DGAttack consistently outperforms DGSlow in transferability attacks from smaller models (e.g., BART) to LLMs like Llama 3.1 and Gemma 2. A key advantage of DGAttack is its black-box nature, which does not depend on model gradients or parameters. This independence reduces the risk of generating adversarial examples that are overfitted or highly specific to the model from which gradients are computed, a common limitation of white-box methods like DGSlow. The results underscore DGAttack's effectiveness and transferability in black-box settings.

Table 3: Comparison of transfer attack results between DGSlow and DGAttack on LLMs. This table shows the performance of adversarial attacks transferred from a smaller model (BART) to LLMs using both DGSlow (white-box) and DGAttack (black-box). **Bold** numbers mean the best metric values across methods.

| Dataset | Method | Llama 3.1 | | | | | | Gemma 2 | | | | | |
|---|---|---|---|---|---|---|---|---|---|---|---|---|---|
| | | GL↑ | BLEU↓ | ROU.↓ | MET.↓ | ASR↑ | Cos.↑ | GL↑ | BLEU↓ | ROU.↓ | MET.↓ | ASR↑ | Cos.↑ |
| BST | DGSlow | 28.34 | 5.33 | 15.00 | 18.53 | 55.97 | **0.92** | 11.63 | 8.20 | 17.47 | 17.73 | 48.61 | **0.92** |
| | DGAttack | **28.38** | **5.20** | **14.70** | **18.23** | **61.03** | 0.81 | **12.72** | **8.13** | **17.03** | **17.37** | **55.70** | 0.81 |
| CV2 | DGSlow | 26.03 | 3.80 | 7.33 | 9.93 | 31.58 | **0.87** | 10.39 | 6.33 | 8.27 | 9.43 | 25.09 | **0.87** |
| | DGAttack | **26.44** | **3.73** | **7.13** | **9.83** | **41.14** | 0.81 | **11.43** | **6.05** | **8.05** | **9.15** | **32.66** | 0.81 |
| PC | DGSlow | 27.38 | 6.40 | 18.53 | 25.13 | 51.14 | **0.89** | 11.27 | 8.73 | 21.80 | 23.50 | 44.57 | **0.89** |
| | DGAttack | **27.74** | **6.23** | **18.17** | **24.87** | **56.32** | 0.82 | **11.42** | **8.50** | **21.47** | **23.00** | **52.80** | 0.82 |
| ED | DGSlow | 26.16 | **3.97** | 7.97 | 9.50 | 49.31 | **0.82** | 12.13 | 6.30 | 9.77 | 9.83 | 43.68 | **0.82** |
| | DGAttack | **26.46** | **3.97** | **7.80** | **9.33** | **54.57** | 0.81 | **12.61** | **6.13** | **9.60** | **9.67** | **49.15** | 0.81 |

In addition to transferability from smaller models, we also compared DGAttack's direct black-box attacks on LLMs with DGSlow's transferability attacks between LLMs (i.e., Llama ↔ Gemma). This comparison, as shown in Table 4, demonstrates that DGAttack, while not outperforming DGSlow's direct white-box attack on LLMs, still performs marginally better than DGSlow's transfer-based attacks between LLMs. The results indicate that DGAttack offers impressive direct attack performance without the need for transfer attack, emphasizing its applicability in real-world scenarios where model-specific knowledge are unavailable.

However, our method has some limitations, particularly in terms of computational and budget constraints, as well as the effectiveness of evolutionary operators. We discuss these challenges, along with future work aimed at addressing them, in Appendix J.

Table 4: Comparison of transfer attacks between LLMs and direct attacks using DGAttack. Rows or columns where the source model and target model are the same (e.g., Llama to Llama) represent direct white-box attacks by DGSlow, and these results are presented in *italics*. **Bold** numbers mean the best metric values across methods.

| Dataset | Method | Llama 3.1 | | | | | | Gemma 2 | | | | | |
|---|---|---|---|---|---|---|---|---|---|---|---|---|---|
| | | GL↑ | BLEU↓ | ROU.↓ | MET.↓ | ASR↑ | Cos.↑ | GL↑ | BLEU↓ | ROU.↓ | MET.↓ | ASR↑ | Cos.↑ |
| BST | Llama | *34.58* | *4.40* | *13.87* | *17.97* | *79.53* | *0.87* | **15.98** | **7.53** | **16.83** | **16.97** | 65.18 | 0.85 |
| | Gemma | **32.23** | 4.77 | 14.37 | 18.20 | 68.35 | **0.85** | *17.87* | *7.17* | *16.50* | *16.67* | *72.92* | *0.87* |
| | DGAttack | 31.90 | **4.67** | **14.13** | **18.17** | 69.32 | 0.86 | 15.96 | 7.57 | 16.83 | 16.97 | 63.67 | **0.86** |
| CV2 | Llama | *32.00* | *3.23* | *6.33* | *9.00* | *64.82* | *0.87* | 13.25 | 5.68 | 7.85 | 8.53 | 51.70 | 0.85 |
| | Gemma | **29.06** | 3.50 | 6.77 | **9.67** | 54.67 | 0.85 | *15.12* | *5.37* | *7.47* | *8.17* | *63.15* | *0.86* |
| | DGAttack | 28.77 | **3.37** | **6.60** | 9.70 | **57.32** | 0.86 | **13.48** | **5.53** | **7.80** | **8.50** | 55.97 | **0.88** |
| PC | LLama | *33.16* | *5.33* | *16.93* | *24.20* | *73.85* | *0.85* | **15.45** | **7.93** | 20.73 | 22.53 | 64.16 | **0.85** |
| | Gemma | **30.33** | 5.77 | 17.53 | **24.30** | 64.54 | 0.85 | *17.06* | *7.60* | *20.07* | *22.13* | *78.36* | *0.86* |
| | DGAttack | 29.24 | **5.67** | **17.50** | **24.30** | 66.05 | 0.84 | 15.20 | **7.93** | **20.60** | **22.37** | 71.11 | 0.84 |
| ED | Llama | *33.22* | *3.47* | *7.57* | *8.93* | *80.79* | *0.85* | **15.93** | 5.77 | 9.40 | 9.43 | 59.28 | **0.86** |
| | Gemma | 30.70 | 3.87 | 7.90 | 9.33 | 67.25 | 0.84 | *17.81* | *5.30* | *9.03* | *9.07* | *77.16* | *0.87* |
| | DGAttack | **30.24** | **3.73** | **7.80** | **9.17** | 71.74 | **0.85** | 15.66 | **5.60** | **9.27** | **9.37** | 62.25 | 0.86 |

## 5 CONCLUSION

In this paper, we proposed DGAttack, a black-box multi-objective attack framework for generating adversarial samples aimed at degrading the performance of dialogue generation models. By leveraging multi-objective evolutionary algorithm (NSGA-II), we simultaneously optimize for two objectives—response length and accuracy. Our method generates adversarial sentences through semantic-preserving perturbations, which ensures that the samples are coherent enough to deceive the dialogue model while substantially reducing the quality of its output. We demonstrate that DGAttack mostly outperforms all black-box, gray-box, white-box baselines and transfer-based white-box attack like DGSlow in black-box settings, particularly against large language models. The ability to directly attack models without relying on access to internal information highlights the practicality and robustness of our approach, proving its applicability in real-world API-based LLM deployments. Our results underscore the power of DGAttack as a state-of-the-art black-box adversarial attack for dialogue generation models, including large-scale models like LLaMA and Gemma.

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

# A  DETAILS OF TARGET MODELS AND DATASETS

## A.1  DATASETS

The statistics for all four datasets are presented in Table 5.

Table 5: Statistics of the four datasets.

| Dataset | #Dialogues | #Utterances |
|---------|-----------|-------------|
| BST | 4,819 | 27,018 |
| PC | 17,878 | 62,442 |
| CV2 | 3,495 | 22,397 |
| ED | 36,660 | 76,673 |

## A.2  TARGET MODELS

Following previous works, we use the HuggingFace pre-trained models—*dialogpt-small*, *bart-base*, and *t5-small* on our main results. For the experiments targeting Large Language Models (LLMs), we employ *gemma-2-9b-it* and *Meta-Llama-3.1-8B-Instruct*. Details of the performance of all victim models are listed in Table 6

Table 6: Performance of five victim models in four benchmark datasets. GL denotes the average generation output length. ROU.(%) and MET.(%) are abbreviations for ROUGE-L and METEOR.

| Dataset | DialoGPT | | | | BART | | | | T5 | | | |
|---------|---------|---------|---------|---------|---------|---------|---------|---------|---------|---------|---------|---------|
| | GL↑ | BLEU↓ | ROU.↓ | MET.↓ | GL↑ | BLEU↓ | ROU.↓ | MET.↓ | GL↑ | BLEU↓ | ROU.↓ | MET.↓ |
| BST | 16.05 | 14.54 | 19.42 | 23.83 | 14.94 | 13.91 | 20.73 | 20.52 | 14.14 | 14.12 | 22.12 | 21.70 |
| CV2 | 12.38 | 12.83 | 16.31 | 14.10 | 10.64 | 12.24 | 11.81 | 12.03 | 13.25 | 10.23 | 10.61 | 9.24 |
| PC | 15.22 | 18.44 | 30.23 | 31.03 | 13.65 | 18.12 | 28.30 | 28.81 | 13.12 | 18.20 | 28.83 | 28.91 |
| ED | 14.47 | 9.24 | 13.10 | 11.42 | 14.69 | 8.04 | 11.13 | 10.92 | 15.20 | 7.73 | 11.31 | 10.34 |

| Dataset | Llama | | | | Gemma | | | |
|---------|---------|---------|---------|---------|---------|---------|---------|---------|
| | GL↑ | BLEU↓ | ROU.↓ | MET.↓ | GL↑ | BLEU↓ | ROU.↓ | MET.↓ |
| BST | 28.10 | 5.40 | 15.27 | 19.13 | 10.78 | 8.33 | 18.03 | 18.20 |
| CV2 | 24.98 | 3.83 | 7.57 | 10.20 | 9.44 | 6.13 | 8.47 | 9.80 |
| PC | 27.27 | 6.33 | 18.57 | 20.22 | 10.41 | 8.73 | 21.80 | 23.10 |
| ED | 23.93 | 4.17 | 8.03 | 9.13 | 10.71 | 6.23 | 9.50 | 9.20 |

## A.3  HYPERPARAMETERS

In our experiments, the minimum similarity threshold $\epsilon$ is set to 0.7 for defining a valid adversarial sentence. For BERT-MLM, we use the HuggingFace pretrained *bert-large-uncased* for masking perturbations given the number of candidates is set to 20. In our Genetic Algorithm implementations, they are installed as only one word within a sentence is perturbed for every generation and the number of generations is set to 5. In other words, there are no more than 5 word-level modifications for every sentence. Following previous work in (Li et al., 2023b), for each dataset, we randomly select 100 dialogue conversations in which each conversation contains 5-8 turns to conduct adversarial attack experiments and evaluate attacking performance.

# B  DGATTACK WITH NON-DOMINATED SORTING GENETIC ALGORITHM II

**Initialization.** DGAttack constructs the initial population $P$ of $N$ candidate adversarial sentences, which are created by randomly perturbing the original input sentence. We evaluate the fitness of each candidate via the two objectives (i.e., AS and GL).

**Binary Tournament Selection.** We create a selection set $S$ containing copies of promising candidate sentences in the current population $P$. Each time, two individuals (i.e., candidate sentences) are randomly sampled from $P$, forming a tournament, and the one with superior fitness (i.e., the one

that Pareto dominates the other) is the winner. If the two individuals are non-dominated with each other, we break the tie randomly. The winner is then cloned into $S$. This process is repeated until the selection set has $N$ selected sentences. Note that we perform sample with replacement so that the current population $P$ remains intact during selection and we allow duplicates in $S$.

**Crossover.** The goal of the crossover operator is to generate new adversarial sentences that inherit beneficial traits from existing sentences. To achieve this, the operator combines segments from each pair of parent sentences in the selection set $S$ to create two new offspring sentences. Let the two parent sentences be $p_1 = (w_1^{(1)}, w_2^{(1)}, \ldots, w_n^{(1)})$ and $p_2 = (w_1^{(2)}, w_2^{(2)}, \ldots, w_n^{(2)})$, where $w_k^{(1)}$ and $w_k^{(2)}$ represent the words in the first and second parent sentences, respectively. A random crossover point $k \in \{1, 2, \ldots, n\}$ is selected. The offspring sentences $o_1$ and $o_2$ are generated by swapping segments from the two parent sentences:

$$o_1 = (w_1^{(1)}, \ldots, w_k^{(1)}, w_{k+1}^{(2)}, \ldots, w_n^{(2)}) \text{ and } o_2 = (w_1^{(2)}, \ldots, w_k^{(2)}, w_{k+1}^{(1)}, \ldots, w_n^{(1)})$$

**Mutation.** The mutation operator introduces random perturbations to selected words within a sentence $p = (w_1, w_2, \ldots, w_n)$. The perturbations should be contextually appropriate, ensuring that the resulting sentences remain coherent and grammatically correct. The mutated sentence $p'$ with its replacement $w_k'$ is represented as: $p' = (w_1, \ldots, w_{k-1}, w_k', w_{k+1}, \ldots, w_n)$

**Non-dominated Sorting.** After variation (i.e., crossover and mutation), we have an offspring set $O$ of $N$ newly-created sentences. We combine both parent and offspring sentences into a pool $(P + O)$ of $2N$ candidates. This pool is then partitioned into non-overlapping subsets $F_i$. Each subset $F_i$, also called a non-dominated sets, contain sentences that are not Pareto dominated by any others in the pool if all subsets of smaller indices $F_1, F_2, \ldots, F_{i-1}$ are removed from the pool. The subset $F_1$ thus contains the best sentences obtained so far as they are not dominated by any other pool members. The subset $F_1$ also forms a non-dominated front in the objective space (GL,AS).

**Replacement.** We need to select $N$ sentences from the pool of $2N$ candidates to form the population for the next generation. Sentences from the non-dominated sets of smaller indices are given priority to be selected first $F_1, F_2, \ldots, F_k$ until $|F_1 \cup F_2 \cup \ldots \cup F_k| \geq N$. We need to select $(N - |F_1 \cup F_2, \cup \ldots \cup F_{k-1}|)$ remaining sentences from $F_k$ based on their crowding distances. This metric measures how far a candidate is from its nearest neighbors of the same non-dominated set in the objective space (GL,AS). Sentences with a higher crowding distance are preferred, as they lie in less populated regions, promoting diversity.

The above procedure of *selection → variation → non-dominated sorting → replacement* is repeated until a termination criterion is satisfied (e.g., reaching the maximum number of generations or running out of the computing budget). Upon termination, the non-dominated set $F_1$ in the population is the approximation set obtained our method. Sentences in the final $F_1$ also forms an approximate non-dominated front in the objective space (GL,AS) that approximates the true Pareto front.

## C  SINGLE-OBJECTIVE GENETIC ALGORITHM

In the single-objective approach, we focus on optimizing one of the following fitness functions to guide the generation of adversarial samples. The fitness function can be designed to either maximize Generation Length or minimize Accuracy Score .

**Fitness Functions**   The fitness functions for the single-objective approach are defined as:

1. **Generation Length:**
$$F_{\text{GL}}(\hat{x}_{B_n}) = \text{GL}(\hat{x}_{B_n}) = |\hat{x}_{A_n}|$$

2. **Accuracy Score:**
$$\text{AS}(\hat{x}_{B_n}) = \text{BLEU}(\hat{x}_{A_n}, x_{A_n}) + \text{ROUGE}(\hat{x}_{A_n}, x_{A_n}) + \text{METEOR}(\hat{x}_{A_n}, x_{A_n})$$

**The optimization process** involves the following steps, as illustrated in Figure  3:

1. **Initialization:** Generate an initial population of candidate adversarial samples by perturbing salient words from the original input sentence.

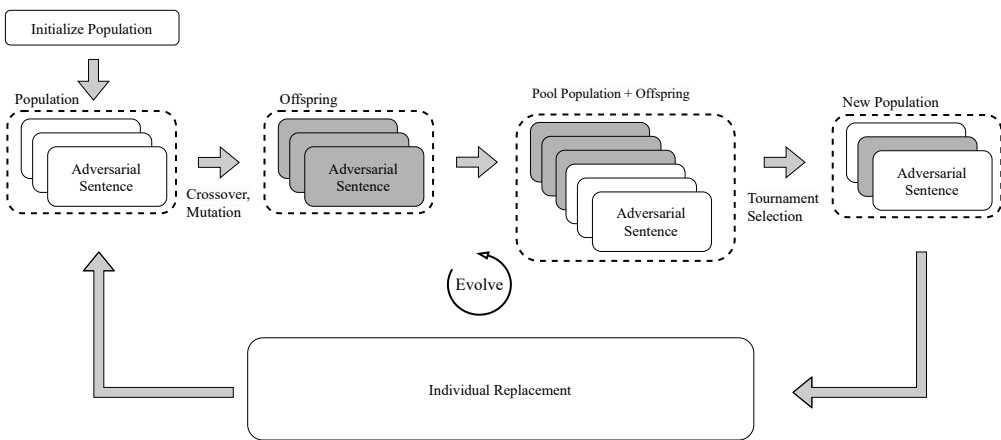

Figure 3: Adversarial attack with Genetic Algorithm

2. **Variation:** Apply mutation and crossover to the current population to create new offspring.

3. **Combination:** Merge the current population with its offspring to form a pool.

4. **Tournament Selection:** Evaluate the fitness of each candidate and select the top-performing individuals within the pool for the next generation.

5. **Iteration:** Repeat the variation and tournament selection steps for a predetermined number of generations.

6. **Final Selection:** After the final generation, sort the candidates in the population based on their fitness scores to choose the most promising candidate for performing the adversarial attack:

   - For Accuracy Score (AS), sort in ascending order to minimize AS.
   - For Generation Length (GL), sort in descending order to maximize GL.

# D COMPUTATIONAL COST COMPARSION

In this section, we provide a detailed comparison of the computational costs associated with the proposed black-box DGAttack method and the baseline white-box DGSlow approach. This analysis focuses specifically on large language models (LLMs) such as Llama 3.1 and Gemma 2, as computational cost considerations are particularly significant for these models due to their high resource demands.

## D.1 OVERVIEW OF COMPUTATIONAL COSTS

We evaluate the runtime and query requirements of both DGSlow and DGAttack under controlled conditions to ensure consistency. All experiments are conducted on a single NVIDIA A100 80GB GPU. Table 7 summarizes the runtime for an entire dataset, the runtime per sample, and the number of queries required per adversarial input.

Table 7: Comparison of computational costs and runtime between DGSlow and DGAttack methods when applied to LLMs such as Llama 3.1 and Gemma 2. The table highlights the presence or absence of gradient access, total runtime for an entire dataset, average runtime per sample, and the number of queries required per adversarial sample. DGAttack incurs higher computational costs due to its black-box nature, which requires iterative evaluation of multiple candidate solutions.

| Attack Method | Gradient Access | Runtime (Dataset) | Runtime (Sample) | Queries/Sample |
|---------------|-----------------|-------------------|------------------|----------------|
| DGSlow | Yes | 8-9 hours | 25-29 seconds | - |
| DGAttack | No | 21-22 hours | 58-66 seconds | 100 queries |

## D.2 RUNTIME ANALYSIS

The runtime comparison reveals that DGAttack incurs approximately 2–2.5x the computational cost of DGSlow when evaluating the same dataset on the same hardware. This increase is primarily due to DGAttack's population-based optimization, which explores a broader adversarial space by iteratively evaluating multiple candidate solutions. However, this trade-off is intrinsic to black-box methods, which must compensate for the lack of access to gradient information by relying on more extensive search strategies.

Through further experimentation, we observe that reducing the population size to 13–15 candidates significantly reduce runtime to approximately 16–18 hours while maintaining high attack effectiveness. This demonstrates that DGAttack can be cost-effective with carefully chosen configurations.

## D.3 QUERY REQUIREMENTS

DGAttack, being a black-box method, requires significantly more queries than DGSlow. For a configuration of 20 candidates and 5 generations, DGAttack necessitates approximately 100 queries per sample. We also conduct experiments with reduced configurations (13–15 candidates), which lower the query requirements to 65–75 per sample while also reducing runtime to 16–18 hours. This finding underscores that DGAttack can achieve cost-efficiency and practicality with well-optimized settings, without significantly compromising attack performance.

# E STANDARD DEVIATIONS FOR REPORTED METRICS

Table 8 reports the standard deviations (std) of metrics (GL, BLEU, ROUGE, METEOR, ASR, Cos) presented in Table 1 across multiple random seeds for DialoGPT, BART, and T5 models across different datasets. Lower std values indicate greater stability, while higher std values may reflect sensitivity to random initialization or dataset-specific variability.

Table 8: Standard Deviations for Results in Table 1

| Dataset | Method | DialoGPT | | | | | | Bart | | | | | | T5 | | | | | |
|---|---|---|---|---|---|---|---|---|---|---|---|---|---|---|---|---|---|---|---|
| | | GL↑ | BLEU↓ | ROU.↓ | MET.↓ | ASR↑ | Cos.↑ | GL↑ | BLEU↓ | ROU.↓ | MET.↓ | ASR↑ | Cos.↑ | GL↑ | BLEU↓ | ROU.↓ | MET.↓ | ASR↑ | Cos.↑ |
| BST | BAE | 1.36 | 0.10 | 0.22 | 0.21 | 1.83 | 0.01 | 1.28 | 0.10 | 0.13 | 0.15 | 1.88 | 0.01 | 0.96 | 0.09 | 0.20 | 0.21 | 2.75 | 0.01 |
| | PWWS | 1.16 | 0.21 | 0.20 | 0.25 | 1.24 | 0.01 | 1.23 | 0.15 | 0.25 | 0.15 | 2.85 | 0.01 | 0.92 | 0.11 | 0.31 | 0.15 | 2.60 | 0.02 |
| | GA(AS) | 2.21 | 0.15 | 0.25 | 0.15 | 2.72 | 0.01 | 1.54 | 0.10 | 0.25 | 0.10 | 1.79 | 0.01 | 1.22 | 0.12 | 0.11 | 0.15 | 2.59 | 0.01 |
| | GA(GL) | 1.41 | 0.10 | 0.21 | 0.21 | 1.84 | 0.02 | 1.24 | 0.15 | 0.15 | 0.20 | 1.90 | 0.01 | 1.14 | 0.10 | 0.20 | 0.15 | 3.79 | 0.01 |
| | DGAttack | 2.53 | 0.20 | 0.14 | 0.12 | 2.60 | 0.01 | 1.26 | 0.15 | 0.23 | 0.25 | 2.41 | 0.02 | 1.72 | 0.12 | 0.30 | 0.36 | 3.67 | 0.03 |
| | DGAttack | 2.29 | 0.15 | 0.17 | 0.15 | 1.97 | 0.01 | 1.34 | 0.21 | 0.24 | 0.24 | 2.35 | 0.02 | 1.28 | 0.10 | 0.31 | 0.26 | 3.19 | 0.01 |
| CV2 | BAE | 1.26 | 0.16 | 0.23 | 0.15 | 1.12 | 0.03 | 1.15 | 0.13 | 0.10 | 0.16 | 2.86 | 0.01 | 1.17 | 0.06 | 0.15 | 0.15 | 1.63 | 0.01 |
| | PWWS | 1.19 | 0.15 | 0.25 | 0.12 | 1.72 | 0.01 | 0.97 | 0.16 | 0.21 | 0.21 | 1.92 | 0.01 | 0.77 | 0.10 | 0.08 | 0.12 | 2.32 | 0.01 |
| | GA(AS) | 1.17 | 0.22 | 0.27 | 0.15 | 1.01 | 0.01 | 1.13 | 0.10 | 0.21 | 0.15 | 2.89 | 0.01 | 0.93 | 0.15 | 0.09 | 0.20 | 2.79 | 0.01 |
| | GA(GL) | 1.24 | 0.10 | 0.16 | 0.13 | 1.95 | 0.01 | 1.17 | 0.15 | 0.15 | 0.20 | 1.84 | 0.03 | 0.95 | 0.15 | 0.10 | 0.15 | 3.64 | 0.01 |
| | DGAttack | 1.47 | 0.34 | 0.41 | 0.27 | 3.20 | 0.01 | 1.61 | 0.25 | 0.26 | 0.23 | 2.54 | 0.01 | 1.27 | 0.32 | 0.14 | 0.31 | 2.68 | 0.02 |
| | DGAttack | 1.17 | 0.18 | 0.35 | 0.25 | 2.31 | 0.02 | 1.51 | 0.15 | 0.19 | 0.28 | 2.75 | 0.01 | 0.84 | 0.12 | 0.18 | 0.10 | 3.06 | 0.01 |
| PC | BAE | 0.88 | 0.15 | 0.25 | 0.45 | 1.14 | 0.01 | 1.16 | 0.16 | 0.10 | 0.26 | 1.57 | 0.02 | 0.64 | 0.15 | 0.15 | 0.21 | 1.83 | 0.03 |
| | PWWS | 0.82 | 0.15 | 0.15 | 0.17 | 1.39 | 0.01 | 2.15 | 0.20 | 0.22 | 0.31 | 1.71 | 0.01 | 0.84 | 0.15 | 0.15 | 0.20 | 2.58 | 0.01 |
| | GA(AS) | 0.91 | 0.15 | 0.15 | 0.20 | 2.78 | 0.03 | 1.56 | 0.15 | 0.15 | 0.24 | 1.73 | 0.02 | 1.17 | 0.21 | 0.21 | 0.15 | 2.65 | 0.02 |
| | GA(GL) | 1.18 | 0.10 | 0.15 | 0.15 | 1.40 | 0.02 | 1.20 | 0.10 | 0.15 | 0.25 | 1.97 | 0.01 | 0.75 | 0.10 | 0.15 | 0.15 | 3.29 | 0.01 |
| | DGAttack | 1.61 | 0.40 | 0.26 | 0.47 | 3.24 | 0.02 | 2.82 | 0.18 | 0.19 | 0.29 | 1.23 | 0.03 | 0.65 | 0.15 | 0.22 | 0.16 | 2.86 | 0.04 |
| | DGAttack | 1.36 | 0.29 | 0.15 | 0.23 | 1.81 | 0.01 | 2.94 | 0.15 | 0.25 | 0.29 | 1.19 | 0.02 | 0.90 | 0.21 | 0.26 | 0.15 | 2.77 | 0.01 |
| ED | BAE | 0.84 | 0.21 | 0.10 | 0.10 | 1.10 | 0.02 | 1.23 | 0.12 | 0.15 | 0.16 | 2.64 | 0.01 | 1.09 | 0.10 | 0.15 | 0.21 | 1.95 | 0.01 |
| | PWWS | 1.21 | 0.12 | 0.16 | 0.12 | 1.80 | 0.01 | 1.82 | 0.17 | 0.15 | 0.21 | 3.20 | 0.03 | 1.06 | 0.15 | 0.10 | 0.18 | 1.87 | 0.04 |
| | GA(AS) | 1.10 | 0.11 | 0.15 | 0.15 | 2.06 | 0.02 | 2.12 | 0.12 | 0.10 | 0.15 | 2.05 | 0.01 | 1.07 | 0.10 | 0.10 | 0.15 | 2.89 | 0.03 |
| | GA(GL) | 1.14 | 0.10 | 0.15 | 0.15 | 2.70 | 0.01 | 2.28 | 0.15 | 0.15 | 0.22 | 2.70 | 0.01 | 1.14 | 0.10 | 0.10 | 0.13 | 4.67 | 0.02 |
| | DGAttack | 1.96 | 0.19 | 0.30 | 0.23 | 2.15 | 0.01 | 2.55 | 0.21 | 0.38 | 0.31 | 3.32 | 0.02 | 1.17 | 0.12 | 0.21 | 0.21 | 3.71 | 0.02 |
| | DGAttack | 1.30 | 0.12 | 0.31 | 0.30 | 1.56 | 0.03 | 2.10 | 0.13 | 0.35 | 0.35 | 2.64 | 0.01 | 1.15 | 0.12 | 0.29 | 0.15 | 3.55 | 0.01 |

Table 9 provides the standard deviations (std) for transferability results on Llama 3.1 8b and Gemma 2 9b. The deviations help assess the robustness of DGAttack against both white-box and transfer attacks, demonstrating its reliability compared to other methods like DGSlow and BART Transfer.

# F CLOSE-SOURCE MODEL EXPERIMENTS

The results in Table 10 illustrate the performance of transferability attacks on the close-source model GPT-4o-mini. The table compares DGAttack's transferability results with those of DGSlow. Notably, the results demonstrate that transfer from DGAttack's black-box samples consistently yields better performance compared to DGSlow's transfer-based attacks.

This finding aligns with observations from our experiments with open-source models. Specifically, DGAttack's ability to generate effective adversarial samples directly, without relying on model-specific knowledge, proves advantageous in scenarios where direct access to the internal workings

Table 9: Standard deviations for results in Tables 3 and 4. Darker-shaded BART rows represent results for DGAttack transferred from BART, while darker-shaded DGAttack rows represent results for DGAttack operating as a direct black-box attack method.

| Dataset | Method | Llama 3.1 8b Instruct | | | | | | Gemma 2 9b it | | | | | |
|---|---|---|---|---|---|---|---|---|---|---|---|---|---|
| | | GL↑ | BLEU↓ | ROU.↓ | MET.↓ | ASR↑ | Cos.↑ | GL↑ | BLEU↓ | ROU.↓ | MET.↓ | ASR↑ | Cos.↑ |
| BST | BART | 2.37 | 0.13 | 0.28 | 0.14 | 1.94 | 0.01 | 1.35 | 0.11 | 0.20 | 0.17 | 5.85 | 0.02 |
| | BART | 1.90 | 0.12 | 0.24 | 0.09 | 5.09 | 0.01 | 1.79 | 0.12 | 0.19 | 0.19 | 5.82 | 0.02 |
| | LLama | 2.63 | 0.22 | 0.29 | 0.26 | 4.70 | 0.02 | 2.50 | 0.26 | 0.26 | 0.19 | 5.65 | 0.01 |
| | Gemma | 1.87 | 0.17 | 0.19 | 0.16 | 1.49 | 0.01 | 2.88 | 0.25 | 0.24 | 0.24 | 6.79 | 0.03 |
| | DGAttack | 1.48 | 0.17 | 0.24 | 0.13 | 3.49 | 0.03 | 2.28 | 0.19 | 0.25 | 0.23 | 4.13 | 0.01 |
| CV2 | BART | 2.34 | 0.08 | 0.26 | 0.21 | 2.03 | 0.03 | 1.28 | 0.05 | 0.38 | 0.34 | 2.44 | 0.01 |
| | BART | 1.54 | 0.09 | 0.33 | 0.29 | 3.20 | 0.01 | 0.91 | 0.05 | 0.35 | 0.35 | 3.33 | 0.01 |
| | LLama | 3.61 | 0.19 | 0.34 | 0.27 | 4.36 | 0.01 | 2.24 | 0.14 | 0.36 | 0.29 | 5.79 | 0.01 |
| | Gemma | 2.89 | 0.14 | 0.35 | 0.25 | 3.47 | 0.02 | 2.26 | 0.17 | 0.26 | 0.30 | 7.26 | 0.02 |
| | DGAttack | 2.82 | 0.17 | 0.23 | 0.27 | 3.51 | 0.00 | 1.78 | 0.12 | 0.37 | 0.29 | 6.12 | 0.00 |
| PC | BART | 1.47 | 0.14 | 0.29 | 0.41 | 2.55 | 0.04 | 1.87 | 0.09 | 0.22 | 0.31 | 2.54 | 0.01 |
| | BART | 2.63 | 0.14 | 0.31 | 0.49 | 2.93 | 0.01 | 1.72 | 0.06 | 0.24 | 0.33 | 2.11 | 0.01 |
| | LLama | 1.32 | 0.21 | 0.35 | 0.37 | 2.82 | 0.01 | 1.69 | 0.12 | 0.35 | 0.26 | 5.52 | 0.01 |
| | Gemma | 2.09 | 0.17 | 0.29 | 0.46 | 3.74 | 0.03 | 2.45 | 0.08 | 0.32 | 0.25 | 2.94 | 0.02 |
| | DGAttack | 1.29 | 0.21 | 0.22 | 0.43 | 2.36 | 0.01 | 1.43 | 0.05 | 0.31 | 0.22 | 4.07 | 0.02 |
| ED | BART | 1.31 | 0.08 | 0.28 | 0.25 | 1.24 | 0.01 | 1.69 | 0.08 | 0.25 | 0.09 | 2.00 | 0.03 |
| | BART | 1.28 | 0.09 | 0.24 | 0.26 | 2.02 | 0.01 | 1.57 | 0.09 | 0.24 | 0.12 | 4.79 | 0.01 |
| | LLama | 3.65 | 0.14 | 0.27 | 0.18 | 2.09 | 0.03 | 2.70 | 0.17 | 0.24 | 0.15 | 5.95 | 0.01 |
| | Gemma | 3.30 | 0.17 | 0.43 | 0.15 | 3.42 | 0.02 | 1.91 | 0.16 | 0.31 | 0.11 | 2.59 | 0.02 |
| | DGAttack | 2.60 | 0.15 | 0.36 | 0.15 | 3.34 | 0.02 | 2.51 | 0.16 | 0.34 | 0.13 | 5.77 | 0.01 |

of the model is not available. This further underscores DGAttack's robustness and practicality in real-world settings.

Table 10: Comparison of transferability between DGSlow and DGAttack on GPT-4o-mini.This table compares transferability attack results of DGSlow and DGAttack on GPT-4o-mini, a close-source model. Dark-shaded rows represent transferability results from DGAttack's black-box samples. **Bold** numbers mean the best metric values across methods.

| Dataset | Method | GPT-4o-mini | | | | | |
|---|---|---|---|---|---|---|
| | | GL↑ | BLEU↓ | ROU.↓ | MET.↓ | ASR↑ | Cos.↑ |
| BST | BART | 16.32 | 10.30 | 19.30 | 21.60 | 26.12 | 0.82 |
| | BART | 16.46 | 10.10 | 18.50 | 21.10 | 30.14 | 0.82 |
| | Llama | 16.52 | **10.00** | 18.10 | 20.70 | 28.06 | **0.85** |
| | Llama | **16.98** | **10.00** | **18.00** | **20.20** | **34.16** | 0.83 |
| | Clean Input | 16.37 | 10.40 | 18.90 | 21.50 | - | - |
| CV2 | BART | 15.39 | 7.40 | 9.80 | 12.50 | 25.36 | 0.80 |
| | BART | 15.44 | 7.30 | 9.40 | 12.50 | 28.48 | 0.81 |
| | Llama | 15.92 | 7.30 | **9.20** | 12.20 | 35.06 | **0.85** |
| | Llama | **15.59** | **7.10** | **9.20** | **12.10** | **37.12** | 0.83 |
| | Clean Input | 15.49 | 7.50 | 10.00 | 12.80 | - | - |
| PC | BART | 16.03 | 11.70 | 23.30 | 29.60 | 35.43 | 0.82 |
| | BART | 16.40 | 11.10 | 23.20 | 29.00 | 36.05 | **0.83** |
| | Llama | **16.62** | **11.20** | 23.10 | 28.90 | 41.06 | 0.81 |
| | Llama | 16.61 | **11.20** | **22.90** | **28.50** | **44.68** | 0.80 |
| | Clean Input | 16.41 | 11.50 | 23.70 | 29.80 | - | - |
| ED | BART | 16.46 | 6.90 | 10.90 | 12.80 | 23.69 | **0.85** |
| | BART | 16.41 | 7.10 | 10.60 | 12.50 | 24.60 | 0.81 |
| | Llama | 16.61 | 6.80 | 10.50 | 12.30 | 28.06 | **0.85** |
| | Llama | **16.69** | **6.50** | **10.30** | **12.20** | **29.22** | 0.83 |
| | Clean Input | 16.52 | 7.00 | 11.20 | 13.10 | - | - |

## G  ETHICS STATEMENT

This work introduces DGAttack, a multi-objective black-box adversarial attack framework designed to evaluate the robustness of dialogue generation (DG) models across four benchmark datasets. The primary aim of this research is to expose vulnerabilities in state-of-the-art DG models, thereby motivating the development of stronger adversarial defenses and more secure systems for real-world applications. By highlighting these vulnerabilities, we hope to raise awareness about potential risks

and inspire the research community to prioritize robustness and security in conversational AI systems.

The ethical implications of this work center around its potential to guide future research toward designing more resilient DG models. Understanding vulnerabilities is a prerequisite for developing effective defenses. DGAttack demonstrates that even black-box methods can significantly compromise DG systems, underscoring the importance of addressing security risks in applications such as virtual assistants, online chatbots, and customer support systems. In alignment with ethical principles established in related works, such as DGSlow, we believe that studying adversarial attacks is a crucial step in improving system resilience and ensuring safer AI deployment.

We acknowledge the dual-use potential of adversarial research, as methodologies designed to reveal system vulnerabilities could also be misused for malicious purposes. However, it is important to emphasize that DGAttack is an untargeted attack. Its primary goal is to disrupt the coherence and consistency of DG models by generating lengthy and irrelevant responses. Unlike targeted attacks, DGAttack does not aim to produce harmful or malicious content, such as offensive or dangerous outputs. This distinction significantly reduces the potential for direct societal harm arising from the misuse of our methodology.

Overall, while research on adversarial attacks carries inherent risks, exposing vulnerabilities in deep learning systems accelerates the development of adversarial defenses. This work contributes to the creation of safer and more reliable AI systems, ensuring their secure deployment in diverse real-world scenarios.

## H ADVERSARIAL DEFENSE AND MITIGATION STRATEGIES

While this work primarily focuses on exposing vulnerabilities in dialogue generation (DG) models through the DGAttack framework, we acknowledge the critical importance of adversarial defenses to mitigate the impact of such attacks. Below, we discuss potential defense mechanisms and strategies that can protect DG systems against adversarial manipulations, thereby aligning with the ethical standards of adversarial machine learning.

### H.1 PROPOSED DEFENSE MECHANISMS

To address the challenges posed by DGAttack and similar adversarial methods, we propose several strategies for mitigating their impact and ensuring the responsible deployment of dialogue generation (DG) systems.

First, adversarial training involves augmenting training datasets with adversarial examples to enhance model robustness by teaching it to handle perturbed inputs Li et al. (2017). Second, input validation and denoising techniques can help detect and mitigate adversarial perturbations before they affect the model, ensuring cleaner inputs Lee & Lee (2018). Third, robust optimization methods, such as regularization techniques and specialized loss functions, can reduce the model's susceptibility to manipulations Zhang et al. (2020a). Lastly, detection pipelines that monitor input-output patterns to flag anomalous behavior indicative of adversarial attacks can serve as an effective defense in deployed systems Nithya et al. (2024).

These strategies, while not implemented or evaluated in this work, are essential for safeguarding DG systems against adversarial threats and ensuring their secure and ethical deployment in real-world scenarios.

### H.2 ALIGNMENT WITH RESEARCH OBJECTIVES

Our primary objective is to reveal vulnerabilities in state-of-the-art DG models, thereby encouraging the development of more secure systems. While we do not implement defense methods in this work, the proposed strategies are intended to stimulate discussions and research on robust defenses. By demonstrating the effectiveness of DGAttack, we aim to motivate further exploration of both attack and defense paradigms, ultimately contributing to the security and reliability of DG systems.

In summary, while this work focuses on exposing vulnerabilities, we emphasize the importance of adversarial defenses in real-world deployments. By encouraging further research in this direction, we aim to ensure the safe and ethical use of DG models in practice.

## I  ABLATION STUDY

We systematically evaluate the impact of various components on the attack efficiency of DGAttack. Specifically, we analyze the effects of the number of generations, the choice of accuracy objective (BLEU, ROUGE, METEOR, or a combined metric), and the influence of the crossover operator.

**The Number of Generations & The Crossover Operator**  The ablation study, as shown in Table 11, examines the influence of the number of generations and the impact of the crossover operator on the final results. Increasing the number of generations leads to a noticeable improvement in attack performance. This outcome is expected, as a greater number of generations allow the algorithm to explore a broader solution space and continuously refine the adversarial samples. With each additional generation, we observe longer outputs (higher GL) and a corresponding degradation in accuracy metrics (BLEU, ROUGE, METEOR). This suggests that more generations enable the adversarial samples to become progressively more disruptive. However, these benefits come at a cost: as the number of generations increases, the cosine similarity between the original and adversarial samples decreases, reflecting the increasing degree of perturbation. While this may contribute to the attack's success, it also indicates that the adversarial samples diverge further from the original content, leading to a trade-off between efficacy and similarity preservation.

Table 11: Ablation study for number of generations and the impact of the crossover operator. **5 Gen Crossover** is the standard method applied in DGAttack, implying that there is no more than 5 changes within a sentence. **Bold** numbers mean the best metric values across methods.

| Dataset | Method | DialoGPT | | | | | | Bart | | | | | | T5 | | | | | |
|---|---|---|---|---|---|---|---|---|---|---|---|---|---|---|---|---|---|---|---|
| | | GL↑ | BLEU↓ | ROU.↓ | MET.↓ | ASR↑ | Cos.↑ | GL↑ | BLEU↓ | ROU.↓ | MET.↓ | ASR↑ | Cos.↑ | GL↑ | BLEU↓ | ROU.↓ | MET.↓ | ASR↑ | Cos.↑ |
| BST | 5 Gen | 21.55 | 13.37 | 18.78 | 22.67 | 50.98 | **0.82** | 29.10 | 8.50 | 17.51 | 22.50 | **74.18** | 0.80 | 20.20 | 10.29 | 19.14 | 21.02 | 60.30 | 0.81 |
| | 10 Gen | **24.20** | **11.60** | 17.91 | 21.88 | 46.57 | 0.78 | 29.52 | **7.75** | 16.89 | 22.10 | 62.50 | 0.74 | 23.48 | 9.97 | **18.62** | 19.54 | 44.12 | 0.75 |
| | 5 Gen Crossover | 22.00 | 12.97 | 19.10 | 22.37 | **52.47** | 0.81 | 28.26 | 8.03 | 17.50 | 22.97 | 70.83 | **0.81** | 19.71 | 10.30 | 18.97 | 20.20 | **69.05** | **0.83** |
| | 10 Gen Crossover | 23.53 | 11.82 | **17.61** | **21.83** | 39.03 | 0.78 | **30.19** | 7.83 | **16.74** | 21.61 | 63.43 | 0.77 | **24.09** | **9.61** | 18.69 | **19.67** | 40.20 | 0.73 |
| CV2 | 5 Gen | 23.09 | 13.61 | 15.21 | 10.18 | 43.11 | **0.84** | 20.14 | 8.30 | 8.58 | 11.23 | **58.24** | 0.83 | 15.27 | 10.17 | 10.40 | 10.70 | 32.30 | 0.80 |
| | 10 Gen | 24.27 | **12.13** | 14.51 | 8.84 | 28.43 | 0.77 | **21.70** | 7.66 | 8.05 | 9.25 | 46.52 | 0.84 | **18.10** | **8.67** | **9.13** | 9.12 | 27.56 | 0.77 |
| | 5 Gen Crossover | 23.94 | 13.27 | 16.43 | 10.73 | **43.74** | 0.80 | 19.78 | 7.93 | 9.13 | 10.93 | 52.99 | 0.81 | 15.57 | 9.93 | 10.27 | 9.80 | **41.22** | **0.82** |
| | 10 Gen Crossover | **24.64** | 12.42 | **13.71** | 9.04 | 29.43 | 0.83 | 21.49 | 7.71 | **7.83** | **9.00** | 43.39 | 0.78 | 17.47 | 8.54 | 9.82 | **9.07** | 30.36 | 0.73 |
| PC | 5 Gen | 20.45 | 17.27 | 28.88 | 29.48 | **53.13** | 0.80 | 25.13 | 10.80 | 23.07 | 30.97 | 66.35 | 0.81 | 18.66 | 12.50 | 26.30 | 28.80 | 45.11 | 0.75 |
| | 10 Gen | 22.13 | 16.69 | 27.48 | 28.27 | 26.22 | 0.72 | **29.05** | 10.60 | 22.77 | 30.90 | 56.41 | 0.76 | 19.33 | 12.27 | 25.80 | 28.60 | 38.41 | 0.71 |
| | 5 Gen Crossover | 19.62 | 17.43 | 28.33 | 28.93 | 48.16 | 0.79 | 25.77 | 10.13 | 22.87 | **30.67** | **66.86** | **0.82** | 18.31 | 12.37 | 26.00 | 28.87 | **50.87** | **0.80** |
| | 10 Gen Crossover | 22.49 | **16.28** | **27.60** | **27.93** | 30.40 | 0.73 | 26.69 | **10.00** | **22.00** | 30.67 | 55.20 | 0.75 | **19.71** | **12.20** | **25.50** | 28.20 | 37.15 | 0.67 |
| ED | 5 Gen | 19.24 | 9.30 | 11.79 | 12.50 | **56.40** | 0.84 | 27.36 | 5.63 | 9.27 | 11.50 | **69.55** | 0.83 | 18.79 | 7.13 | 11.07 | 11.10 | 59.17 | 0.79 |
| | 10 Gen | 20.17 | **8.71** | 11.39 | 10.51 | 38.52 | 0.77 | 29.46 | 5.17 | 8.87 | **10.60** | 62.11 | 0.76 | **20.31** | 6.50 | 9.81 | 9.93 | 38.22 | 0.73 |
| | 5 Gen Crossover | 19.80 | 9.43 | 11.67 | 11.80 | 48.91 | 0.81 | 27.68 | 5.27 | 9.13 | 11.57 | 69.22 | 0.81 | 18.53 | 7.07 | 10.37 | 10.47 | **63.11** | **0.82** |
| | 10 Gen Crossover | 19.89 | 8.74 | **11.30** | **10.47** | 34.06 | 0.75 | **29.61** | **5.13** | **8.85** | 11.03 | 52.47 | 0.75 | 19.84 | **6.42** | **9.73** | 9.97 | 40.13 | 0.71 |

As for the crossover operator, we observe relatively little difference between the performance with and without crossover, indicating that the single-point crossover used here may be too simple or straightforward to offer significant benefits in this context. This suggests that while crossover helps introduce variation in traditional genetic algorithms, it may not be as critical for generating effective adversarial samples in our scenario. More sophisticated crossover methods or higher complexity operators could potentially yield different results, but in this case, the simplicity of the single-point crossover did not contribute substantial advantages.

**The Choice of Accuracy Objective**  The ablation study presented in Table 12 explores the effect of using different performance metrics (BLEU, ROUGE, and METEOR) as the accuracy objective for minimization. In our main experiments, we aggregated all three metrics to form a combined accuracy objective, which was used to guide the adversarial attack. Interestingly, the results indicate that when comparing the combined objective to individual metrics, the performance differences were marginal. This implies that while each metric focuses on distinct aspects of accuracy—BLEU emphasizing n-gram precision, ROUGE measuring recall, and METEOR accounting for semantic similarities through synonyms and paraphrasing—their roles in contributing to the degradation of the generated text are largely aligned.

One notable observation is that minimizing any single accuracy metric often triggers a reduction in the others as well. For example, when focusing solely on minimizing BLEU, we see that ROUGE and METEOR scores also tend to degrade. This suggests that there is a degree of overlap in the dimensions these metrics assess. BLEU's focus on exact matches between the generated and reference text often overlaps with ROUGE's focus on recall (how much of the reference is captured

by the generation), and METEOR's consideration of paraphrasing and synonyms further ties into this. Consequently, degradation in one metric is likely to cause a cascading effect, pulling the others down in tandem.

This cascading degradation across metrics highlights an important insight: adversarial samples crafted to minimize a single accuracy metric are likely to be effective in degrading the overall quality of the generated text. This occurs because each metric, though distinct, evaluates overlapping characteristics of fluency, coherence, and relevance. Thus, regardless of whether BLEU, ROUGE, or METEOR is targeted directly, the adversarial attack tends to degrade performance across all three metrics to some extent.

That being said, the combined accuracy objective remains the most holistic approach. By aggregating BLEU, ROUGE, and METEOR into a single metric, the adversarial attack is forced to address all facets of text quality simultaneously—precision, recall, and semantic similarity. This makes the attack stronger and ensures a comprehensive degradation of the generated responses. While optimizing for a single metric may still result in an effective attack, the combined approach ensures a more robust and consistent reduction in overall text quality across all dimensions, leading to a more impactful attack outcome.

Table 12: Ablation study for the choice of accuracy objectives. **COMBINED** is the accuracy score (AS) applied in DGAttack. **Bold** numbers mean the best metric values across methods.

| Dataset | Method | DialoGPT | | | | | | Bart | | | | | | T5 | | | | | |
|---|---|---|---|---|---|---|---|---|---|---|---|---|---|---|---|---|---|---|---|
| | | GL↑ | BLEU↓ | ROU.↓ | MET.↓ | ASR↑ | Cos.↑ | GL↑ | BLEU↓ | ROU.↓ | MET.↓ | ASR↑ | Cos.↑ | GL↑ | BLEU↓ | ROU.↓ | MET.↓ | ASR↑ | Cos.↑ |
| BST | BLEU | 21.01 | 12.97 | 19.57 | 22.60 | 48.68 | **0.82** | 28.24 | 8.23 | 17.60 | **22.77** | 71.48 | **0.82** | 20.54 | 10.95 | 19.53 | 20.97 | 56.86 | 0.79 |
| | ROUGE | 22.86 | **12.50** | 19.93 | **21.50** | 45.06 | 0.81 | 28.11 | 8.37 | 17.90 | 22.95 | 67.43 | 0.81 | 19.49 | 11.15 | 20.53 | 20.93 | 64.96 | **0.81** |
| | COMBINED | 22.00 | 12.97 | **19.10** | 22.37 | **52.47** | 0.81 | 28.26 | **8.03** | **17.50** | 22.97 | 70.83 | 0.81 | 19.71 | **10.30** | **18.97** | **20.20** | **69.05** | 0.80 |
| | METEOR | **22.89** | 13.60 | 20.80 | 22.70 | 49.02 | **0.82** | **28.53** | 8.15 | 17.67 | 22.87 | 68.53 | 0.81 | 20.49 | 10.43 | 20.50 | 20.83 | 51.41 | 0.79 |
| CV2 | BLEU | 22.34 | 13.24 | **16.01** | 9.97 | 41.94 | **0.84** | 19.78 | 8.00 | 9.10 | 11.10 | **59.80** | **0.82** | 15.94 | 11.03 | **10.20** | 10.47 | 46.71 | 0.80 |
| | ROUGE | 23.23 | 13.00 | 16.23 | 9.80 | **52.47** | 0.80 | **20.79** | **7.77** | 8.83 | 10.80 | 53.29 | 0.79 | 15.76 | 10.67 | 10.53 | **9.70** | 40.43 | 0.81 |
| | COMBINED | **23.94** | 13.27 | 16.43 | 10.73 | 43.74 | 0.80 | 19.78 | 7.93 | 9.13 | 10.93 | 52.99 | **0.82** | 15.57 | **9.93** | 10.27 | 9.80 | 41.22 | **0.82** |
| | METEOR | 23.43 | **12.53** | 16.48 | **9.71** | 50.22 | 0.81 | 20.72 | 7.80 | **8.63** | **10.33** | 52.90 | **0.82** | **16.42** | 10.67 | 10.47 | 9.80 | **47.35** | 0.79 |
| PC | BLEU | 18.05 | 19.27 | 28.73 | 30.70 | **51.00** | 0.80 | 24.92 | 10.57 | 22.87 | **30.33** | 64.78 | 0.81 | 19.16 | 13.27 | 28.17 | 28.63 | 42.94 | **0.82** |
| | ROUGE | **19.93** | 18.30 | 29.70 | **28.90** | 48.34 | **0.82** | 25.30 | 10.37 | 22.50 | 30.37 | 59.11 | **0.83** | 18.18 | 14.77 | 28.25 | **28.40** | 46.66 | **0.82** |
| | COMBINED | 19.62 | **17.43** | **28.33** | 28.93 | 48.16 | 0.79 | **25.77** | 10.13 | 22.87 | 30.67 | **66.86** | 0.82 | 18.31 | **12.37** | **26.13** | 28.87 | **50.87** | 0.80 |
| | METEOR | 19.87 | 18.13 | 28.80 | 29.40 | 44.50 | 0.81 | 25.30 | **10.07** | **22.13** | **30.33** | 56.52 | 0.81 | **19.29** | 14.63 | 27.50 | **28.40** | 50.15 | 0.81 |
| ED | BLEU | 19.16 | 8.70 | **11.53** | 12.13 | **51.52** | **0.84** | 27.87 | **5.17** | 9.70 | 12.33 | 67.82 | 0.80 | **18.58** | 7.50 | 10.83 | 11.03 | 50.32 | 0.81 |
| | ROUGE | 19.64 | 8.82 | 11.57 | 11.57 | 46.98 | 0.81 | **28.95** | 5.27 | 9.97 | 13.27 | 60.72 | **0.82** | 18.16 | 7.47 | 11.30 | **10.27** | 50.43 | **0.83** |
| | COMBINED | **19.80** | 9.43 | 11.67 | 11.80 | 48.91 | 0.81 | 27.68 | 5.27 | 9.13 | 11.57 | **69.22** | 0.81 | 18.53 | **7.07** | **10.37** | 10.47 | **63.11** | 0.82 |
| | METEOR | 19.78 | **8.60** | 11.77 | **11.17** | 48.74 | 0.82 | 27.52 | 5.20 | **8.80** | **10.43** | 67.96 | **0.82** | 17.66 | 7.30 | 11.53 | 11.50 | 50.71 | 0.81 |

## J LIMITATIONS

Our method still remains several limitations as listed:

**Mutation.** We use POS tags to identify salient words within a sentence for masking and word substitution with BERT. An effective heuristic integrated into our mutation operator could better select important words for substitution, leading to higher-quality candidates.

**Crossover.** To avoid errors such as breaking word linkages and grammatical mistakes during crossover, our operator is relatively simple and straightforward, only swapping each segment from two sentences. A more complex and efficient operator could enhance the diversity among candidates and improve the attack success rate.

**Attacking LLMs.** Large Language Models (Brown et al., 2020) (LLMs) are highly robust due to extensive training on diverse datasets. They can effectively handle minor word-level substitutions, making small perturbations is insufficient for effective attacks. More sophisticated strategies are required to challenge LLMs and degrade their performance.

**Trade-offs and Computational Considerations.** While our black-box method shows promising results in attacking LLMs, it comes with increased computational costs. As evolutionary algorithms require evaluating numerous candidate solutions, leading to longer attack times and high-computational cost. In contrast, gradient-based methods like DGSlow are generally faster due to the direct use of gradient information. This highlights a broader limitation of our empirical evaluation, which, due to computational and budget constraints, was confined to specific datasets, formats, and models. While our method has demonstrated notable results on smaller models, these constraints may limit the generalizability of our findings, particularly for LLMs. Expanding future experiments to include larger-scale datasets, more diverse formats, and additional task categories could provide further insights into the broader applicability of our approach. In scenarios where demonstrating vulnerabilities in LLMs is critical, the additional computational effort may be justified.

