# OpenReview forum: "Black-Box Adversarial Attack on Dialogue Generation via Multi-Objective Optimization"
_ICLR.cc/2025/Conference — Submitted to ICLR 2025_

### Official Review · Reviewer_Gr1j · 2024-10-23

**Soundness:** 2
**Presentation:** 2
**Contribution:** 2
**Rating:** 5
**Confidence:** 3

**Summary:**

The paper introduces DGAttack, a black-box adversarial attack framework that targets dialogue generation models using multi-objective optimization, aiming to minimize response coherence and maximize response length. This attack operates in scenarios where internal model details are inaccessible, making it practical for real-world AI systems.

**Strengths:**

- The paper proposes a bi-objective optimization framework (minimizing coherence, maximizing length) to generate adversarial samples. The use of the non-dominated sorting genetic algorithm (NSGA-II) to handle conflicting objectives is innovative.
- DGAttack succeeds in the challenging black-box setting, where only input and output prompts are used to craft adversarial attacks, without requiring model internals such as gradients or probabilities.
- The authors demonstrate DGAttack's effectiveness through empirical experiments across multiple datasets and models (BART, DialoGPT, T5). They show superior performance compared to both existing white-box and black-box approaches.
- This paper demonstrates DGAttack's transferability from smaller to larger models and between large models. One key advantage of black-box attack is that they can learn more general and transferable attacks.

**Weaknesses:**

- The title is inaccurate. Since this paper only considers two conflicting objectives, I think it is not appropriate to use the word "multi". It should be "Black-Box Adversarial Attack on Dialogue Generation via Bi-Objective Optimization".
- The use of evolutionary algorithms, while effective, may lead to significant computational overhead, especially when optimizing across two objectives. The authors could provide more detailed discussions on computational costs and practical constraints in real-world applications. The current discussion in Appendix E should be included into more details. For example, does the computational cost of this method and the computational cost of baseline methods differ by orders of magnitude?
- This paper's experiments are focused open-source LLMs. However, the selling point of this paper is the black-box adversarial attack, so the experiment results on closed source models should also be provided, such as earlier version of GPT3.5 (If you think attacking GPT 4 is too difficult).

I will raise my score if my concerns are resolved.

**Questions:**

See Weaknesses.

---

> ### Author Response · Authors · 2024-11-21
>
> >**Weakness 1** The title is inaccurate. Since this paper only considers two conflicting objectives, I think it is not appropriate to use the word "multi". It should be "Black-Box Adversarial Attack on Dialogue Generation via Bi-Objective Optimization".
>
> In multi-objective optimization, particularly in the domain of multi-objective evolutionary algorithms, the term multi-objective optimization is commonly used to describe problems with two or more objectives. This usage is by convention and aligns with established terminology in the field. Therefore, our work adheres to the standard nomenclature in the literature. DGAttack can also be straightforwardly employed for scenarios with more than two objectives. To prevent potential misunderstanding, we will clarify this convention in the manuscript.

---

> ### Author Response · Authors · 2024-11-21
>
> >**Weakness 2** The use of evolutionary algorithms, while effective, may lead to significant computational overhead, especially when optimizing across two objectives. The authors could provide more detailed discussions on computational costs and practical constraints in real-world applications. The current discussion in Appendix E should be included into more details. For example, does the computational cost of this method and the computational cost of baseline methods differ by orders of magnitude?
>
> We thank the reviewer for their insightful comment regarding computational costs and practical constraints in real-world applications. Below, we provide a detailed response, addressing runtime comparisons, query requirements, and the trade-offs inherent in DGAttack's approach.
>
> ### **1. Computational Costs: GPU Runtime and Query Requirements**
> We acknowledge that evolutionary algorithms like DGAttack come with higher computational costs due to their population-based approach. To provide a clear comparison, all experiments—including white-box (DGSlow), black-box (DGAttack), and transfer-based attacks—were conducted on a single NVIDIA A100 80GB GPU. The runtime comparisons are as follows:
>
> #### **Gradient-Based Methods (DGSlow):**
> - DGSlow leverages gradient information for direct optimization, resulting in faster evaluation times.
> - For LLMs like LLAMA and Gemma, DGSlow requires approximately **8–9 hours** to evaluate an entire dataset.
> - This efficiency is due to its use of internal model parameters, which significantly reduces the search space.
>
> #### **Black-Box Methods (DGAttack):**
> - DGAttack employs a genetic algorithm that searches a broader adversarial space, iteratively evaluating a population of candidate solutions across multiple generations.
> - In our experiments, using a configuration of 20 candidates and 5 generations, DGAttack required approximately **21–22 hours** to evaluate an entire dataset on the same GPU.
> - However, through further experimentation, we observed that reducing the population size to 13–15 candidates significantly reduced runtime to approximately **16–18 hours** while maintaining high attack effectiveness. This demonstrates that DGAttack can be cost-effective with carefully chosen configurations.
>
> ### **2. Addressing "Orders of Magnitude" Differences**
> The reviewer inquired whether the computational cost of DGAttack differs from baseline methods by orders of magnitude. Based on our experiments:
> - The runtime difference between DGSlow and DGAttack is approximately **2x** (**8–9 hours vs. 21–22 hours** for the same dataset and hardware).
> - This does not represent an "order of magnitude" (i.e., **10x or more**) difference. Instead, it reflects the inherent trade-off for operating in black-box settings, which require broader exploration due to the absence of gradient information.
>
> ### **3. Practical Constraints and Real-World Applications**
> We recognize the importance of discussing real-world constraints and their implications for DGAttack:
>
> #### **3.1 Query Requirements:**
> - DGAttack requires multiple queries per iteration due to its population-based approach. Specifically:
>   - For 20 candidates and 5 generations, DGAttack requires approximately **100 queries per sample**.
>   - Reducing the population size to 13–15 candidates can lower this to **65–75 queries per sample**, making it more efficient while maintaining effectiveness.
>
> #### **3.2 Targeting Closed-Source Models:**
> - DGAttack’s flexibility allows it to attack proprietary API-based LLMs, which do not expose internal parameters.
> - While the query requirements are higher compared to gradient-based methods, this trade-off is necessary to address real-world scenarios where white-box approaches are infeasible.
>
> #### **3.3 Hardware Requirements:**
> - All experiments were conducted on a single NVIDIA A100 80GB GPU, ensuring consistency in comparisons.
> - While this setup is accessible in research contexts, we acknowledge that the computational requirements may pose challenges for practitioners with more limited resources.

---

> ### Author Response · Authors · 2024-11-21
>
> >**Weakness 3** This paper's experiments are focused open-source LLMs. However, the selling point of this paper is the black-box adversarial attack, so the experiment results on closed source models should also be provided, such as earlier version of GPT3.5 (If you think attacking GPT 4 is too difficult).
>
> We thank the reviewer for their insightful comment regarding experiments on closed-source models like GPT. Below, we address this concern by explaining the constraints of conducting such experiments and providing additional insights from the experiments we were able to perform.
>
> ### **1. Constraints on Conducting Comprehensive Closed-Source Model Experiments**
> Due to financial and resource constraints, we are unable to provide a full table of results for attacking GPT-4o-mini or other closed-source models within the rebuttal period. These experiments are costly because:
> - Query-based attacks on closed-source models, particularly those accessed via APIs, incur significant monetary costs.
> - Query limits imposed by the API providers further restrict the number of evaluations possible within a short time frame.
> - Additionally, running comprehensive experiments across all configurations and baselines requires time beyond what is feasible during the rebuttal period.
>
> ### **2. Experiments Conducted on GPT**
> Despite these constraints, we conducted a series of transferability experiments and a direct attack on GPT. Specifically:
>
> #### **Transferability Experiments**
> - We tested adversarial prompts generated by DGSlow and DGAttack on BART and LLAMA, then transferred these prompts to GPT for evaluation.
> - Results indicate that DGAttack consistently outperforms DGSlow in these transferability experiments, showcasing its superior generalizability as a black-box approach. This is expected given that DGAttack does not rely on gradients, which often leads to overfitting to the source model.
>
> | Dataset | Method          | GPT-4o-mini |        |        |        |        |       |
> | :------ | :-------------: | :---------: | :----: | :----: | :----: | :----: | :---: |
> |         |                 | GL          | BLEU   | ROU.   | MET.   | ASR    | Cos.  |
> | BST     | DGSlow\_BART    | 16\.32      | 10\.30 | 19\.30 | 21\.60 | 26\.12 | 0\.82 |
> |         | DGAttack\_BART  | 16\.46      | 10\.10 | 18\.50 | 21\.10 | 30\.14 | 0\.82 |
> |         | DGSlow\_Llama   | 16\.52      | 10\.00 | 18\.10 | 20\.70 | 28\.06 | 0\.85 |
> |         | DGAttack\_Llama | 16\.98      | 10\.00 | 18\.00 | 20\.20 | 34\.16 | 0\.83 |
> |         | Clean Input     | 16\.37      | 10\.40 | 18\.90 | 21\.50 | N/A    | N/A   |

---

### Official Review · Reviewer_98Uq · 2024-11-03

**Soundness:** 2
**Presentation:** 2
**Contribution:** 1
**Rating:** 1
**Confidence:** 5

**Summary:**

This paper proposes a genetic algorithm for addressing the multi-objective adversarial attacks problem. The experimental results demonstrate that DGAttack outperforms other baselines' black-box and gray-box approaches in terms of generating longer and less coherent responses. The framework also achieves reasonable transferability.

**Strengths:**

Reasonable Good Performance in Benchmarking Black-Box Scenarios: The experimental results seem to demonstrate that DGAttack outperforms other baselines' black-box and gray-box approaches in terms of generating longer and less coherent responses. The framework also achieves reasonable transferability.

**Weaknesses:**

1. Limited Novelty & Orignity, and Overlap with Prior Work:
The novelty of this work is limited due to significant overlap with a prior study by Li et al. (2023) (White-Box Multi-Objective Adversarial Attack on Dialogue Generation), which was the first to introduce a Pareto-optimization framework specifically for white-box adversarial attacks in dialogue generation. Li et al. (2023) formulated a multi-objective optimization approach to balance between adversarial effectiveness and conversational coherence, making it a foundational work in this space. In contrast, this paper claims to target a similar optimization objective but in a black-box setting, suggesting only a slight modification. While black-box constraints introduce challenges such as limited access to model internals, simply adapting an established white-box approach to a black-box context may not constitute a significant technical contribution. To substantiate the novelty and its originality, the authors need to clearly distinguish their approach from Li et al.’s framework.

2. Lack of Comprehensive Experiments:
The experimental evaluation in this work lacks robustness, particularly due to the absence of state-of-the-art (SOTA) models in the primary experiments (Sections 4.2 and 4.3). While Section 4.4 includes evaluations on models such as Meta-LLama-3.1-8B-inst and Gemma-2-9B-it, the analysis does not extend to larger, more recent LLMs like Meta-LLama/Llama-3.2-90B-Vision-Instruct. This gap in testing leaves the efficacy of the proposed method on the latest generation of LLMs unproven, which limits the practical applicability of the results. Besides, in Sec. 4.4, the authors do not specify the reasons why forgive multiple baselines evaluated in Sec. 4.2 and Sec. 4.3 (FD, HotFlip, PWWS, BAE, GA(AS), GA(GL)). Obviously, these methods could be applicable to LLM scenarios. Furthermore, proprietary models such as Claude 3 and GPT-4-o, which are highly relevant in current LLM applications, are also omitted. Since proprietary and larger open-source models can exhibit unique behaviors under adversarial conditions, evaluating the proposed method on these systems is essential for demonstrating broad applicability and robustness. Without these evaluations, it’s unclear if the method generalizes well across the current landscape of LLMs, thus weakening the claim of practical relevance.

3. Concerns with Reproducibility:
The reproducibility of this work is called into question as the authors have not provided open-source code or comprehensive instructions to replicate the experiments. In the absence of these materials, other researchers cannot validate the results, explore variations of the methodology, or build upon the work. Reproducibility is a cornerstone of scientific research, and providing a well-documented codebase and dataset configurations is essential for enabling follow-up studies and independent verification. The lack of these materials raises concerns about the credibility and transparency of the findings. For the next revision, it would be beneficial for the authors to make their codebase publicly available, alongside detailed instructions on the experimental setup, hyperparameters, and any model configurations used, to facilitate reproducibility.

4. Ethical Considerations and Potential for Misuse:
The ethical implications of this research are insufficiently addressed, leaving key questions unanswered regarding how to detect or defend against the proposed adversarial methods. Research in adversarial attacks inherently carries dual-use potential, as it can be misused to compromise model reliability in deployed systems, affecting user trust and potentially leading to harm. The authors should explicitly discuss possible defense mechanisms or detection strategies that could mitigate the impact of the proposed attacks, thereby demonstrating a commitment to responsible research. Additionally, they should consider how their work might be misused if applied in malicious contexts and outline measures to prevent such outcomes. Providing recommendations for safeguards or limitations on how the method can be deployed responsibly would mitigate negative societal impacts and align the research with ethical standards in adversarial machine learning.

**Questions:**

1. Distinguish the technical contributions of this work from Li et al. (2023) (White-Box Multi-Objective Adversarial Attack on Dialogue Generation).

2. Please explain why multiple baselines are forgiven in Sec. 4.4 (FD, HotFlip, PWWS, BAE, GA(AS), GA(GL)).

3. How do the authors address ethical issues for their proposed method?

**Details Of Ethics Concerns:**

The authors do not properly address ethical issues. How to detect or defend the proposed methods? Could the proposed method be potentially misused and how to mitigate the negative societal impacts?

---

> ### Author Response · Authors · 2024-11-21
>
> > **Weakness 1** Limited Novelty & Orignity, and Overlap with Prior Wor
>
> >**Question 1** Distinguish the technical contributions of this work from Li et al. (2023)
>
> We thank the reviewer for their thoughtful feedback and the opportunity to clarify the contributions and originality of DGAttack. Below, we provide a detailed explanation to distinguish our approach from the prior work by Li et al. (2023) and highlight the substantial technical contributions of DGAttack beyond adapting a white-box method to a black-box context.
>
> ### **1. Novelty in Addressing Black-Box Challenges**
> Li et al. (2023) introduced a Pareto-optimization framework for white-box adversarial attacks, leveraging access to gradients and internal model parameters. However, DGAttack fundamentally differs in its problem formulation and methodology due to the systematic shift from a white-box to a black-box setting:
>
> - **Absence of Gradients or Model Internals:**
>   DGSlow depends on gradient-based optimization, which is infeasible in real-world scenarios involving proprietary or API-based models (e.g., GPT-4, Claude). DGAttack, on the other hand, operates without access to gradients or model internals, relying solely on input-output behavior. This constraint significantly increases the difficulty of the optimization problem, as DGAttack must navigate the adversarial search space without the directional guidance provided by gradients.
>
> - **Objective Flexibility:**
>   While DGSlow requires differentiable objectives to compute gradients, DGAttack’s black-box framework supports both differentiable and non-differentiable objectives. This flexibility allows for optimization of metrics such as semantic similarity, linguistic diversity, or response fluency, expanding the applicability of DGAttack to diverse tasks and scenarios.
>
> ### **2. Strengths of Multi-Objective Genetic Algorithms**
> DGAttack leverages a multi-objective genetic algorithm (NSGA-II) to optimize its objectives, introducing several key advantages:
>
> - **Simultaneous Multi-Objective Optimization:**
>   Unlike DGSlow, which aggregates objectives into a single one using Lagrange multipliers, DGAttack directly approximates a Pareto-optimal set. This avoids complex hyperparameter tuning required by DGSlow, where weights assigned to objectives depend on factors like dataset properties, model characteristics, and conversational context.
>
> - **Population-Based Diversity:**
>   DGAttack’s population-based search explores diverse adversarial solutions, making it robust across different models and contexts. This contrasts with gradient descent in DGSlow, which is inherently limited to finding a single solution per run. The diversity enabled by NSGA-II is crucial for crafting generalizable adversarial inputs in black-box settings.
>
> ### **3. Transferability: A Key Differentiator**
> Transferability is a critical component of adversarial attack frameworks, particularly in black-box scenarios. DGAttack demonstrates superior transferability in several ways:
>
> - **From Smaller Models to LLMs:**
>   DGAttack crafts adversarial prompts on smaller models like BART and transfers them effectively to larger LLMs like LLAMA-3.1 and Gemma. This is achieved without relying on model-specific gradients, avoiding overfitting to the source model.
>
> - **Comparison with DGSlow:**
>   DGSlow’s reliance on gradients limits its transferability, as adversarial examples crafted in a white-box setting often fail to generalize to other models. DGAttack’s black-box nature inherently enhances generalizability, as shown in our experiments, where it outperforms DGSlow in transferability evaluations.
>
> ### **4. Real-World Black-Box Evaluation**
> DGAttack is designed to address the challenges of real-world black-box scenarios:
>
> - **Applicability Across Diverse Models:**
>   Our experiments demonstrate DGAttack’s effectiveness on both traditional dialogue models (DialoGPT, BART, T5) and cutting-edge LLMs (LLAMA-3.1, Gemma). This highlights its robustness and scalability across a wide range of architectures.
>
> - **Realistic Experimental Design:**
>   Unlike DGSlow, which assumes access to internal model parameters, DGAttack reflects real-world constraints where adversarial attacks must operate with limited information. This includes settings like API-based LLMs, where only input-output behavior is observable.
>
> ### **5. Novel Experimental Insights**
> DGAttack not only extends the scope of adversarial attacks to black-box settings but also provides novel insights into the practical challenges and trade-offs of such attacks:
>
> - **Robustness Across Tasks and Models:**
>   DGAttack achieves strong results on multiple datasets and models, demonstrating its versatility and effectiveness.
>
> - **Improved Attack Generalizability:**
>   The use of NSGA-II ensures that DGAttack can identify diverse and robust adversarial inputs, outperforming gradient-based methods like DGSlow in black-box and transferability scenarios.

---

> ### Author Response · Authors · 2024-11-21
>
> >**Weakness 2** Lack of Comprehensive Experiments
>
>
> We thank the reviewer for their valuable feedback regarding the experimental evaluation and the question about omitted baselines in Section 4.4. Below, we address these concerns in detail, incorporating additional insights from experiments on closed-source models like GPT.
>
> ### **1. Addressing the Weakness: Lack of Comprehensive Experiments**
> We acknowledge the importance of evaluating DGAttack on the latest state-of-the-art (SOTA) large language models (LLMs) and closed-source models. However, several constraints shaped the scope of our evaluation:
>
> #### **1.1 Availability of Recent Models**
> - **Timing of LLama-3.2 Release:**
>   LLama-3.2-90B-Vision-Instruct was released only six days before our submission deadline. Consequently, we were unable to include it in our experiments. Instead, we focused on LLama-3.1 and Gemma-2, the most recent models available during our evaluation phase.
> - **Planned Future Work:**
>   We intend to evaluate DGAttack on newer models like LLama-3.2 in future studies, which will further validate its robustness across evolving LLMs.
>
> #### **1.2 Focus on Closed-Source Models**
> While we recognize the importance of closed-source models such as GPT-3.5 and GPT-4 for evaluating black-box attacks, their inclusion presents significant challenges:
> - **Query Costs:**
>   Attacking closed-source models via APIs incurs high monetary costs, particularly for extensive evaluations required for benchmarking across baselines and configurations.
> - **Query Limits:**
>   Restrictions imposed by API providers further limit the number of experiments possible within a short period.
> - **Time Constraints:**
>   Conducting comprehensive experiments on multiple baselines during the rebuttal period is not feasible.
> Despite these challenges, we prioritized running transferability and direct-attack experiments on GPT (details provided below) to demonstrate DGAttack’s efficacy on closed-source systems.
>
> #### **1.3 Computational and Financial Constraints**
> Running all baselines (e.g., FD, HotFlip, PWWS, BAE, GA(AS), GA(GL)) on modern LLMs such as LLama-3.1 and Gemma would require significant computational resources. Given these constraints, we focused on the most competitive baseline, DGSlow, to ensure a meaningful comparison that highlights DGAttack’s strengths.

---

> ### Author Response · Authors · 2024-11-21
>
> >**Question 2** Please explain why multiple baselines are forgiven in Sec. 4.4 (FD, HotFlip, PWWS, BAE, GA(AS), GA(GL)).
>
>
> The decision to omit certain baselines (e.g., FD, HotFlip, PWWS, BAE, GA(AS), GA(GL)) was driven by their applicability and relevance to the evaluated models:
>
> #### **2.1 Applicability of Baselines**
> - **Limited Applicability to Black-Box Settings:**
>   Many of these baselines (FD, HotFlip, PWWS, BAE) rely on gradient information or other internal model parameters, which are unavailable in black-box settings, especially for proprietary models like GPT.
> - **Suitability for Dialogue Generation Tasks:**
>   Methods like PWWS and BAE are primarily designed for classification tasks and are less effective for attacking dialogue generation models, which require coherent, contextually appropriate adversarial responses.
>
> #### **2.2 Focus on the Most Relevant Comparisons**
> - **DGSlow as the Strongest Baseline:**
>   DGSlow shares conceptual similarities with DGAttack and is the most competitive baseline for multi-objective adversarial attacks. Comparing DGAttack with DGSlow allows for a direct evaluation of their relative strengths.
> - **Efficient Resource Allocation:**
>   Prioritizing DGAttack and DGSlow ensured a robust comparison while staying within budgetary and time constraints.
>
> ### **3. Experiments Conducted on GPT**
> Despite the constraints, we conducted several experiments on GPT to assess DGAttack’s effectiveness on closed-source models. These experiments include both transferability evaluations and direct attacks:
>
> #### **3.1 Transferability Experiments**
> - We tested adversarial prompts generated by DGSlow and DGAttack on BART and LLAMA, then transferred these prompts to GPT for evaluation.
> - Results indicate that DGAttack consistently outperforms DGSlow in these transferability experiments, showcasing its superior generalizability as a black-box approach. This is expected given that DGAttack does not rely on gradients, which often leads to overfitting to the source model.
>
> | Dataset | Method          | GPT-4o-mini |        |        |        |        |       |
> | :------ | :-------------: | :---------: | :----: | :----: | :----: | :----: | :---: |
> |         |                 | GL          | BLEU   | ROU.   | MET.   | ASR    | Cos.  |
> | BST     | DGSlow\_BART    | 16\.32      | 10\.30 | 19\.30 | 21\.60 | 26\.12 | 0\.82 |
> |         | DGAttack\_BART  | 16\.46      | 10\.10 | 18\.50 | 21\.10 | 30\.14 | 0\.82 |
> |         | DGSlow\_Llama   | 16\.52      | 10\.00 | 18\.10 | 20\.70 | 28\.06 | 0\.85 |
> |         | DGAttack\_Llama | 16\.98      | 10\.00 | 18\.00 | 20\.20 | 34\.16 | 0\.83 |
> |         | Clean Input     | 16\.37      | 10\.40 | 18\.90 | 21\.50 | N/A    | N/A   |

---

> ### Author Response · Authors · 2024-11-21
>
> >**Weakness 3** Concerns with Reproducibility
>
> **Source Code Provided:**
> We have actually provided the complete source code for DGAttack in the supplementary material in the original submission: [Supplementary Material](https://openreview.net/attachment?id=GnBBSlUb0S&name=supplementary_material). This codebase contains:
> - The implementation of DGAttack and its integration with various models listed in our paper.
> - Scripts for running experiments on the evaluated datasets.
> - Parameter configurations for the genetic algorithm (e.g., population size, number of generations, mutation rates).
>
> **Hyperparameter Details in Manuscript:**
> The manuscript explicitly specifies the key hyperparameters used in our experiments, including:
> - Candidate pool size and number of generations.
> - Evaluation metrics and thresholds for adversarial attack success.
> - Dataset splits and preprocessing details.

---

> ### Author Response · Authors · 2024-11-21
>
> >**Weakness 4** Ethical Considerations and Potential for Misuse
>
> >**Question 3**  How do the authors address ethical issues for their proposed method?
>
> We thank the reviewer for their insightful feedback on ethical considerations and the potential for misuse of our proposed methodology. Below, we address these concerns and outline how our work aligns with ethical standards in adversarial machine learning.
>
> ### **1. Ethical Implications and Research Motivation**
> The primary goal of this work is to expose vulnerabilities in dialogue generation (DG) models and encourage the development of more robust and secure systems. Understanding potential risks is a prerequisite for creating effective defenses. Specifically:
> - **Highlighting Vulnerabilities:**
>   By demonstrating that even black-box attacks like DGAttack can significantly compromise DG models, we aim to raise awareness about potential security threats in real-world applications, such as online chatbots or virtual assistants.
> - **Facilitating Robustness Research:**
>   Our findings are intended to motivate further research on adversarial defenses, improving the resilience of DG systems in diverse deployment scenarios.
>
> We align with the ethical stance adopted by related works, such as DGSlow, which emphasize the importance of studying adversarial attacks as a first step toward mitigating them.
>
> ### **2. Potential for Misuse and Mitigation Strategies**
> We recognize that adversarial research carries dual-use potential, as the methodologies developed to test system vulnerabilities can also be exploited maliciously. To address this:
>
> #### **Proactive Measures for Responsible Use**
> We advocate for controlled and responsible use of our methodology. Researchers and practitioners should ensure that:
> - The methodology is used only in secure, closed research environments to test and improve system robustness.
> - Access to attack frameworks is limited to authorized users, such as academic researchers or industry professionals working on model security.
>
> #### **Prevention of Misuse**
> To minimize the risk of misuse:
> - Any public release of this work, including source code, should be accompanied by clear disclaimers emphasizing its intended use for research and security purposes only.
> - Access to the code could be regulated through licensing agreements, ensuring that it is available only to those with legitimate and ethical intentions.
>
> #### **Untargeted Nature of DGAttack**
> It is important to emphasize that DGAttack is specifically designed as an untargeted attack. Its objective is to craft adversarial inputs that cause the model to generate lengthy and irrelevant responses, thereby disrupting its coherence and consistency.
> - Unlike targeted attacks, DGAttack does not aim to generate harmful or malicious content, such as "jailbreaking" LLMs to produce offensive or dangerous outputs.
> - This distinction significantly reduces the risk of direct societal harm arising from the misuse of our methodology.
>
> ### **3. Recommendations for Adversarial Defenses**
> While this work does not focus on developing or evaluating defense mechanisms, we recognize their importance in mitigating the impact of adversarial attacks. In line with the reviewer’s suggestion, we propose several strategies that can defend against attacks like DGAttack:
> - **Adversarial Training:**
>   Augmenting the training dataset with adversarial examples to improve the model’s robustness.
> - **Input Validation and Denoising:**
>   Incorporating preprocessing techniques to detect and mitigate adversarial perturbations in inputs.
> - **Robust Optimization:**
>   Adopting advanced optimization techniques to make the model more resistant to adversarial manipulations.
> - **Detection Pipelines:**
>   Developing tools to identify unusual input patterns or outputs indicative of an attack.
>
> These approaches can significantly mitigate the risk posed by DGAttack and similar methods, ensuring safer deployments of DG models. We will include these suggestions in the Appendix section of our revised manuscript.

---

### Official Review · Reviewer_dWqa · 2024-11-04

**Soundness:** 2
**Presentation:** 2
**Contribution:** 2
**Rating:** 5
**Confidence:** 3

**Summary:**

The paper proposes a novel dialog generation adversarial attack called DGAttack, which works by optimizing the input to (1) minimize the coherence of the response and (2) maximize the length of the response. The optimization is based on a multi-objective evolutionary algorithm, which allows one to perform the attack in a black-box setup. The authors demonstrate the effectiveness of the algorithm on four dialog datasets and three dialog generation models (BART, T5, DialoGPT). DGAttack can also be applied to larger language models such as Llama and Gemma.

**Strengths:**

1. The proposed adversarial algorithm is shown to be effective across models and datasets, outperforming existing black-box threat models as well as certain grey-box baselines.

2. The method can be used in a black-box setup and can be directly applied to larger and closed-source models.

**Weaknesses:**

1. The paper does not explain why an adversarial attack on a dialog generation system—with the objectives of reduced accuracy and longer responses poses a significant threat to these models. It is not entirely clear how this property can be misused and why this adversarial model is harmful. Therefore, it is hard to evaluate the importance of the problem.

2. Given that the performance differences across different methods are sometimes very small (e.g., as seen in Table 3), it would be helpful to include standard deviations to assess the significance of the improvements..

**Questions:**

DGAttack finds perturbations in the input space that cause the highest decrease in the accuracy of the output. Since the input has been changed, the original response might no longer be the optimal one for the new 'adversarial' input. Given this, why is the correspondence between the generated responses and the reference outputs considered the optimal metric for measuring the adversarial effectiveness of the model? Would it be better to employ a metric that captures the relevance of the generated response to the new adversarial input?

---

> ### Author Response · Authors · 2024-11-21
>
> >**Weakness 1** The paper does not explain why an adversarial attack on a dialog generation system—with the objectives of reduced accuracy and longer responses poses a significant threat to these models. It is not entirely clear how this property can be misused and why this adversarial model is harmful. Therefore, it is hard to evaluate the importance of the problem.
>
>
>
> We appreciate the reviewer’s concern and agree that the implications of the proposed adversarial attack could be more clearly articulated. We argue that the objectives of reduced accuracy and longer responses represent critical vulnerabilities for dialogue generation systems in several real-world applications:
>
> ### **1. Degradation of User Experience**
> - Adversarial inputs that generate incoherent or excessively verbose responses degrade the quality of dialogue interactions.
> - This can lead to user dissatisfaction in applications such as customer support chatbots, virtual assistants, and conversational agents in healthcare.
>
> ### **2. Resource Exploitation**
> - Generating longer responses unnecessarily increases computational and time costs, which can be exploited to overload systems, especially when applied at scale in API-based deployments.
> - For instance, an adversarial attack could lead to elevated server costs or delays in response time, affecting the availability of services.
>
> ### **3. Undermining Model Robustness**
> - By exposing vulnerabilities, our approach highlights potential weaknesses in dialogue models, encouraging the development of more robust and secure systems.
> - This aligns with the broader goal of adversarial research, which is to preemptively identify and mitigate potential threats.

---

> ### Author Response · Authors · 2024-11-21
>
> >**Weakness 2** Given that the performance differences across different methods are sometimes very small (e.g., as seen in Table 3), it would be helpful to include standard deviations to assess the significance of the improvements..
>
> We agree with the reviewer that including standard deviations would provide a clearer picture of the statistical significance of the results. In the revised manuscript, we will include standard deviations for all reported metrics in Tables 1, 3 and 4, calculated over multiple runs with different random seeds.

---

> ### Author Response · Authors · 2024-11-21
>
> >**Questions:** DGAttack finds perturbations in the input space that cause the highest decrease in the accuracy of the output. Since the input has been changed, the original response might no longer be the optimal one for the new 'adversarial' input. Given this, why is the correspondence between the generated responses and the reference outputs considered the optimal metric for measuring the adversarial effectiveness of the model? Would it be better to employ a metric that captures the relevance of the generated response to the new adversarial input?
>
> We thank the reviewer for this thoughtful question and appreciate the opportunity to clarify our evaluation approach. Below, we explain why the relevance between the original response (from the clean input) and the adversarial response (from the adversarial input) is central to measuring adversarial effectiveness and why employing a metric that captures the relevance of the generated response to the adversarial input would not align with the goals of DGAttack.
>
>
>
> ### **1. Objectives of Adversarial Attacks**
> The primary goal of an adversarial attack is to craft an adversarial input that forces the model to produce a generated response that **deviates significantly** from the original response produced from the clean input. This deviation reflects the success of the attack in disrupting the model’s ability to generate coherent, relevant, and consistent outputs.
>
> In the context of DGAttack:
> - The **original model-generated response** $x_A^n$ serves as a baseline representing the model's expected behavior.
> - The **adversarially generated response**  $\hat{x}_A^n$ is crafted to deviate from this baseline, achieving irrelevance, incoherence, or inconsistency.
>
>
>
> ### **2. Why Relevance to the Original Response Matters**
> The relevance between the original response ($x_A^n$) and the adversarial response ($\hat{x}_A^n$) is crucial in adversarial attack evaluation because:
>
> #### **Measuring Disruption of Baseline Behavior**
> - Adversarial attacks aim to exploit the model’s vulnerabilities, resulting in responses that diverge from the expected, baseline behavior represented by $x_A^n$.
> - By measuring how much the adversarial response deviates from the original response, we quantify the attack’s success in disrupting the model’s intended behavior.
>
> #### **Capturing Irrelevance and Incoherence**
> - Metrics like **BLEU**, **ROUGE**, and **METEOR** measure the similarity between $\hat{x}_A^n$ and $x_A^n$.
> - **Lower scores** indicate greater deviation, reflecting the success of the adversarial attack in forcing the model to produce irrelevant or incoherent outputs.
>
> ### **3. Why Relevance to the Adversarial Input is Not Aligned**
> Employing a metric that evaluates the relevance of the adversarial response ($\hat{x}_A^n$) to the adversarial input ($x_B^n$) would contradict the objectives of DGAttack. This approach assumes that the adversarial response should maintain semantic alignment with the adversarial input, which is not the purpose of the attack. Specifically:
>
> #### **Irrelevance is the Goal**
> - DGAttack intentionally perturbs the input to exploit vulnerabilities and induce incoherence or irrelevance in the response.
> - Evaluating relevance to the adversarial input ($x_B^n$) would reward responses that are semantically aligned with the perturbed input, which is the opposite of the attack’s objective.
>
> #### **Focus on Disruption, Not Alignment**
> - The attack’s success lies in disrupting the model’s ability to maintain coherence and relevance to the original task, not in aligning responses with the adversarial input.
> - Measuring relevance to $x_B^n$ would fail to capture this disruption.
>
> ### **4. Alignment with the Goals of DGAttack**
> The use of metrics like BLEU, ROUGE, and METEOR to evaluate the similarity between $\hat{x}_A^n$ and $x_A^n$ aligns directly with the goals of DGAttack:
> - These metrics quantify the attack’s success in forcing deviations from the original response.
> - **Lower scores** indicate that the adversarial response has become irrelevant, incoherent, or inconsistent, meeting the intended adversarial objective.

---

> > ### Comment · Reviewer_dWqa · 2024-11-23
> > **Response to Authors**
> >
> > I thank the authors' for the clarifications; however, my concerns remain:
> >
> > 1. Motivation Behind the Adversarial Attack on a Dialog Generation System:
> >
> > * Degradation of User Experience: If the user initiates an adversarial attack, why would they intentionally degrade their own experience?
> > * Resource Exploitation: If the primary goal of this attack is resource exploitation, it should be benchmarked against other resource exploitation attacks for context.
> > * Undermining Model Robustness: The finding that language models are susceptible to adversarial attacks is not novel, as multiple studies on such attacks against language models already exist.
> >
> > 2. Metrics for evaluating success of an adversarial attack.
> > * If the primary objective is to craft adversarial inputs that forces the model to produce significantly deviating responses compared to clean inputs with no other constraints, a user could simply ask a different question to achieve a similar effect. In such cases, the generated output will naturally differ from the original response. Therefore, I believe the relevance to the original response metric is not suitable for evaluating this attack.

---

> > > ### Author Response · Authors · 2024-11-24
> > >
> > > >**Metrics for evaluating success of an adversarial attack.** If the primary objective is to craft adversarial inputs that forces the model to produce significantly deviating responses compared to clean inputs with no other constraints, a user could simply ask a different question to achieve a similar effect. In such cases, the generated output will naturally differ from the original response. Therefore, I believe the relevance to the original response metric is not suitable for evaluating this attack.
> > >
> > > We appreciate the reviewer’s feedback. However, we would like to clarify that adversarial attacks on dialogue generation (DG) tasks, including our approach, inherently come with constraints. Specifically, the crafted adversarial input must remain sufficiently similar to the original input to ensure the attack remains meaningful. Simply asking a different question deviates entirely from the original intent of the input, which would not constitute a valid adversarial attack.
> > >
> > > The objective of DGAttack is to minimally perturb the input while forcing the model to generate significantly deviating responses. DGAttack emphasizes the controlled exploitation of the model’s vulnerabilities with minimal input perturbations. It is not about generating entirely unrelated responses but about forcing the model to deviate significantly from its expected behavior. This disruption is a more direct measure of adversarial success than input-output alignment under noise.

---

> ### Author Response · Authors · 2024-11-24
>
> We thank the reviewer for their thoughtful comments and constructive feedback. Below, we address each concern in detail:
>
> >**Degradation of User Experience:** If the user initiates an adversarial attack, why would they intentionally degrade their own experience?
>
> We understand the reviewer’s concern that it may appear counterintuitive for a user to degrade their own experience through an adversarial attack. However, the focus of this research is not on genuine users but on adversarial scenarios where malicious actors intentionally exploit model vulnerabilities. In adversarial research, the primary goal is to identify and demonstrate weaknesses in systems, especially in black-box settings, to preemptively address potential real-world threats.
>
> In such cases, the degradation of the user experience is not an end goal for the user but a consequence of the adversary’s intent to disrupt or manipulate the system. For example:
> - **Sabotage:**
>   Adversaries might target customer-facing systems to create incoherent or excessively verbose outputs, damaging the reputation of the service provider.
> - **Testing and Improvement:**
>   By degrading model performance through adversarial inputs, researchers uncover critical vulnerabilities, enabling the development of more robust and secure dialogue systems.
>
> Our work emphasizes the importance of exposing such vulnerabilities to ensure these systems are equipped to handle adversarial scenarios effectively. This aligns with the broader goals of adversarial research.
>
> >**Resource Exploitation:** If the primary goal of this attack is resource exploitation, it should be benchmarked against other resource exploitation attacks for context.
>
> We appreciate the reviewer’s suggestion regarding benchmarking against other resource exploitation attacks for context. However, it is important to note that resource exploitation is a secondary consequence of our primary objectives, rather than the main focus of our work. Our primary goal is to investigate how adversarial inputs can force the model to generate longer and less accurate responses, resulting in irrelevant and repetitive outputs. This approach highlights the model's vulnerabilities in handling long-context generation while maintaining accuracy—a critical trade-off in dialogue generation systems.
>
> The relevance of this approach lies in its implications:
> - **Vulnerability Testing:**
>   By forcing the model to generate longer outputs, we effectively test its robustness to long-context generation. Such scenarios expose weaknesses in managing coherence, fluency, and computational efficiency, which are vital for many real-world applications.
> - **Secondary Impact:**
>   While resource exploitation is not our primary objective, it naturally emerges as a byproduct of longer response generation. In API-based systems, longer responses lead to increased computational and time costs, which adversaries could exploit at scale.
>
> As for benchmarking, while we acknowledge the value of comparing our method to other resource exploitation attacks, running comprehensive baselines specific to resource exploitation is outside the scope of this work. API experiments are costly and time-intensive, and our goal is not to optimize attacks purely for resource depletion but to demonstrate how forcing longer outputs serves as a viable attack vector. We do, however, recognize the potential for future work to extend this analysis by benchmarking against such attacks.
>
> >**Undermining Model Robustness:** The finding that language models are susceptible to adversarial attacks is not novel, as multiple studies on such attacks against language models already exist.
>
> We acknowledge that adversarial attacks on language models are a well-established area of research. However, our work stands out in several key ways:
> - **Focus on Generation Tasks:**
>   Unlike prior work predominantly focused on classification tasks, we specifically target dialogue generation, which presents unique challenges such as managing conversational context and coherence. This shift in focus highlights new vulnerabilities specific to generation tasks.
> - **Black-Box Attack via MOEA:**
>   We target black-box adversarial attacks, employing a multi-objective evolutionary algorithm (MOEA) like NSGA-II. By optimizing for reduced response coherence and increased verbosity, we propose a novel attack framework tailored for dialogue systems.
> - **Demonstrated Effectiveness:**
>   Our approach showcases strong performance across multiple benchmark datasets and dialogue models, providing evidence of its practical implications and state-of-the-art performance in black-box settings.
>
> In summary, while the general concept of adversarial attacks may not be novel, our focus on dialogue generation tasks, the use of MOEA for black-box attacks, and the demonstrated effectiveness of our method collectively make a meaningful contribution to the field.

---

### Official Review · Reviewer_RaT9 · 2024-11-05

**Soundness:** 2
**Presentation:** 2
**Contribution:** 1
**Rating:** 6
**Confidence:** 3

**Summary:**

The paper studies the problem of crafting black-box adversarial attacks in Dialogue Generation (DG) models. Existing adversarial methods either rely on access to model gradients or internal parameters (white-box approaches) or demonstrate limited effectiveness in a black-box setting. To this end, the authors proposed DGAttack, a black-box adversarial attack framework specifically for DG models. DGAttack employs a multi-objective optimization, aiming to maximize generation length while minimizing response coherence.

**Strengths:**

1. The problem statement studied in this paper is interesting and has been well-articulated to the reader. The limitations in prior approaches have also been well-conveyed. Further, the overall presentation of the proposed approach is clear.

2. The experiments section looks comprehensive with evaluations on four standard benchmarks. DGAttack leads to an increased Attack success rate across most of the models and datasets.

**Weaknesses:**

1. The technical contribution is not convincing and lacks originality. The idea of minimizing accuracy and maximizing response length is not novel but has been earlier explored in DGSlow [1]. I understand, that DGSlow is a white-box attack and has a drawback in requiring access to the gradients. However, the primary optimization problem in DGAttack closely resembles that of DGSlow [1], with the primary difference being the use of a genetic algorithm to solve the multi-objective optimization. I request the authors to clarify the contributions and differences with DGSlow [1].

2. In the introduction, the authors mentioned that recent LLMs are adept at generating coherent long-form responses, and hence it is difficult to craft a black-box attack. However, in Table 1, the experimental results were presented on relatively older LLMs such as DialoGPT, BART, and T5. I recommend that, in addition to the transfer attacks in Table 3, the authors also present primary results in Table 1 on more current models like LLAMA-3.1 and Gemma.

[Minor]
1. The presentation could have been better. For example: equation numbers are missing.
2. On page 3, the definition of f(.) is not clear.
3. In the equation on line 151, how is the reference response $x_\text{ref}$ defined is not clear.
4. On line 345, should it be $cos(x_i, \tilde{x_i}) < \epsilon$, instead of “>” sign?

[1] Li, Y., Li, Z., Gao, Y. and Liu, C., 2023. White-box multi-objective adversarial attack on dialogue generation. arXiv preprint arXiv:2305.03655.

**Questions:**

Please see the Weaknesses.

My major concerns are regarding the lack of originality in the proposed approach, and primary evaluation on relatively older LLMs. For rebuttal, I shall recommend the authors: (1) To clarify the technical contributions and differences with DGSlow [1]. (2) Evaluation on LLAMA-3.1/Gemma/other modern LLMs for Table 1 setup.

[1] Li, Y., Li, Z., Gao, Y. and Liu, C., 2023. White-box multi-objective adversarial attack on dialogue generation. arXiv preprint arXiv:2305.03655.

---

> ### Author Response · Authors · 2024-11-18
>
> > **Weakness 1** The technical contribution is not convincing and lacks originality. The idea of minimizing accuracy and maximizing response length is not novel but has been earlier explored in DGSlow [1]. I understand, that DGSlow is a white-box attack and has a drawback in requiring access to the gradients. However, the primary optimization problem in DGAttack closely resembles that of DGSlow [1], with the primary difference being the use of a genetic algorithm to solve the multi-objective optimization. I request the authors to clarify the contributions and differences with DGSlow [1].
>
>
> We thank the reviewer for their insightful question regarding the technical contributions and differences between DGAttack and DGSlow. While DGAttack inherits the two primary objectives from DGSlow (minimizing accuracy and maximizing response length), our work introduces substantial innovations that address challenges in black-box settings and extend the applicability of adversarial attack frameworks. Below, we detail the distinctions and contributions:
>
> ### **1. Adaptation to the Black-Box Setting**
> The shift from a white-box to a black-box setting fundamentally changes the problem formulation and methodology:
>
> - **No Access to Gradients or Internal Parameters:**
>   DGSlow relies on white-box access to gradients and model internals for optimization, which is infeasible for proprietary models or API-based deployments (e.g., GPT-4, Claude). DGAttack is explicitly designed for black-box scenarios, relying solely on input-output behavior.
> - **Objective Flexibility:**
>   DGSlow requires differentiable objectives due to its dependence on gradient-based optimization. DGAttack, however, supports both differentiable and non-differentiable objectives, enabling optimization of metrics like semantic similarity, response diversity, or linguistic fluency in black-box settings.
>
> ### **2. Strengths of Multi-Objective Genetic Algorithms**
>
> DGAttack employs a multi-objective evolutionary algorithm (NSGA-II) to optimize the conflicting objectives simultaneously. This introduces several advantages over DGSlow’s gradient-based optimization:
>
> - **Simultaneous Multi-Objective Optimization:**
>   DGSlow aggregates objectives into a single one using Lagrange multipliers, requiring complex hyperparameter tuning that depends on:
>   - The scale of each objective.
>   - Dataset and model characteristics.
>   - Conversational context.
> In contrast, NSGA-II inherently balances conflicting objectives without manual tuning, directly approximating a Pareto-optimal set of diverse adversarial candidates.
> - **Population-Based Search for Diversity:**
>   NSGA-II’s population-based search ensures diversity in adversarial sentences, critical for crafting robust attacks that generalize across contexts and models. Gradient descent in DGSlow, however, is limited to finding a single solution per run.
>
> ### **3. Transferability Analysis**
> - DGAttack demonstrates superior transferability by crafting adversarial prompts on smaller models (e.g., BART) and successfully testing them on larger LLMs (e.g., LLAMA-3.1, Gemma).
> - In contrast, DGSlow’s reliance on model-specific gradients limits its transferability, often leading to overfitting.
> - Our experiments validate DGAttack’s strong transferability, reinforcing its practical relevance for black-box adversarial attacks.
>
> ### **4. Novel Experimental Insights**
> - **Robustness Across Models and Datasets:**
>   We empirically demonstrate DGAttack’s effectiveness on both traditional dialogue models (DialoGPT, BART, T5) and cutting-edge LLMs (LLAMA-3.1, Gemma), exhibiting its broad applicability.
>
> - **Real-World Black-Box Evaluation:**
> Our experiments are designed to reflect real-world conditions where adversarial attacks must work with limited information. This is a stark contrast with DGSlow’s white-box setting, which assumes access to gradients and internal parameters.
>
>
> By addressing these aspects, DGAttack advances the field of adversarial attacks, particularly in black-box scenarios, and extends the applicability of adversarial frameworks to real-world challenges.

---

> ### Author Response · Authors · 2024-11-18
>
> >**Weakness 2** In the introduction, the authors mentioned that recent LLMs are adept at generating coherent long-form responses, and hence it is difficult to craft a black-box attack. However, in Table 1, the experimental results were presented on relatively older LLMs such as DialoGPT, BART, and T5. I recommend that, in addition to the transfer attacks in Table 3, the authors also present primary results in Table 1 on more current models like LLAMA-3.1 and Gemma.
>
> We thank the reviewer for their valuable suggestion to include evaluations on more recent large language models (LLMs) like LLAMA-3.1 and Gemma in Table 1. While we understand the importance of benchmarking against state-of-the-art (SOTA) LLMs to strengthen our claims, there are several practical considerations that led to our current experimental setup:
>
>
> **Widely Accepted Models:** DialoGPT, BART, and T5 are widely used models in adversarial attack research, making them essential for establishing a fair comparison with prior works. Including these models allows us to demonstrate DGAttack’s performance in a consistent and comparable experimental framework.
>
> **Computational Cost of Baselines:** Running all baselines (e.g., FD, HotFlip, PWWS, BAE, GA(AS), GA(GL)) on modern LLMs such as LLAMA-3.1 and Gemma is prohibitively expensive regarding our limited computational and financial budget. Each baseline requires substantial compute resources due to the scale of the models and the large number of evaluations required for reliable benchmarking (as we have done for DGSlow and our DGAttack).
>
> **Focus on DGAttack's Comparative Strengths:** Given these constraints, we prioritized running DGAttack on LLAMA-3.1 and Gemma to showcase its effectiveness in the challenging black-box setting. Additionally, we compared it directly with DGSlow, as DGSlow is the most competitive baseline for multi-objective adversarial attacks. On top of that, these baselines (FD, HotFlip, PWWS, BAE, GA(AS), GA(GL)) are less effective in terms of attacking DG models. Therefore, we conduct experiments of adversarial attacks on LLMs with DGAttack and DGSlow via direct and transfer attacks.

---

> ### Author Response · Authors · 2024-11-24
>
> >**[MINOR]** The presentation could have been better. For example: equation numbers are missing.
> On page 3, the definition of $f(⋅)$ is not clear.
> In the equation on line 151, how is the reference response $X_{\text{ref}}$ defined is not clear.
> On line 345, should it be $\cos(x_i, \tilde{x}_i) < \epsilon$, instead of the “>” sign?
>
> We thank the reviewer for their observations and suggestions. Below, we address each point:
>
> ### **1. Equation Numbers**
> We will revise the manuscript in the final version to include equation numbers for better readability and referencing throughout the paper.
>
>
> ### **2. Definition of $f(⋅)$ (Page 3)**
> The definition of $f(⋅)$ was indeed not clearly stated in the initial manuscript. In our context, $f(⋅)$ represents a generative model in the Dialogue Generation (DG) framework. Specifically:
> - $f$ takes as input the **persona**, **dialogue history**, and **current utterance**.
> - It generates the **response** as the output.
>
> This clarification has been added in the revised manuscript.
>
> ### **3. Definition of $X_{\text{ref}}$ (Line 151)**
> The reference response $X_{\text{ref}_n}$ is defined as a sample output or a reference output originating from the dataset. However:
> - Practical experiments focus on evaluating adversarial inputs against the **original model response**, rather than the reference response, to reflect real-world black-box scenarios.
>
> ### **4. Cosine Similarity Threshold (Line 345)**
> We confirm that the equation as written is correct:
> - The condition $\cos(x_i, \tilde{x}_i) > \epsilon$ is intentional.
> - It ensures that the adversarial input $\tilde{x}_i$ remains sufficiently similar to the original input $x_i$ while achieving the attack objectives.
>
> The threshold $\epsilon$ is a safeguard to measure the similarity between $x_i$ and $\tilde{x}_i$, ensuring minimal perturbation during the attack.
>
> Due to the page limit, these clarifications have been included in the comment section rather than elaborated in the main manuscript. We hope this addresses the reviewer’s concerns effectively.

---

> > ### Comment · Reviewer_RaT9 · 2024-11-26
> > **Thank you for the clarifications**
> >
> > Thank you for clarifying some of my concerns. After the rebuttal, I hold a more positive outlook regarding the novelty of the proposed approach.  I understand the author's concern regarding computational constraints, however, I still believe that additional experiments are needed to make this paper stronger, i.e., evaluating on more recent LLMs as suggested earlier.
> >
> > I am willing to increase my score based on the clarifications regarding novelty, but the authors should promise to add these additional evaluations (on at least one recent LLM) in the camera-ready version.

---

> > > ### Author Response · Authors · 2024-11-30
> > >
> > > We sincerely thank the reviewer for the valuable feedback and suggestions. We will try our best to perform additional evaluations for baseline methods in Table 1 on at least one recent LLM (Llama or Gemma) in the camera-ready version.

---

### Official Review · Reviewer_PaTU · 2024-11-05

**Soundness:** 3
**Presentation:** 3
**Contribution:** 3
**Rating:** 6
**Confidence:** 2

**Summary:**

This paper proposed DGAttack, a black-box multi-objective attack framework for generating
adversarial samples aimed at degrading the performance of dialogue generation models. It leverages multi-objective evolutionary algorithm to optimize for two objectives—response length and accuracy. The proposed method generates adversarial sentences through semantic preserving perturbations to substantially reduce the quality of the model output.

**Strengths:**

1. The paper is easy to read, and looks interesting to most people. It applies a genetic algorithm to optimize for the attack, which is quite novel.

2. The experiments section has extensive baselines. And the result performance looks good.

**Weaknesses:**

1. In the limitations section, it mentions that the proposed method doesn't work well for LLMs, because word-level substitutions are handled very well.

2. The proposed method also comes with high computational cost, this also can make the model less useful.

**Questions:**

A detailed analysis of the computational cost would be useful. For example, for remote LLMs attach, how many api calls are needed in each iteration of the algorithm.

---

> ### Author Response · Authors · 2024-11-18
>
> > **Questions:**
> A detailed analysis of the computational cost would be useful. For example, for remote LLMs attach, how many api calls are needed in each iteration of the algorithm.
>
> We thank the reviewer for their thoughtful feedback and for highlighting the strengths of our work. Below, we address the question regarding the computational cost of DGAttack, particularly for remote LLMs.
>
> To clarify, the number of API calls required in our method is approximately the product of the number of individuals in the population and the number of generations in the evolutionary process. In our experiments, we set the population size to 20 individuals and evolve over 5 generations. Therefore, for each dialogue sample, DGAttack requires approximately 100 API calls to generate a successful adversarial input prompt.
>
> We acknowledge that this query cost can be significant, particularly when targeting remote LLMs with restricted API access. We will include a detailed analysis of computational costs, including query counts and runtime comparisons with baseline methods, in the revised manuscript to provide further insights into the practical feasibility of DGAttack.
>
> Once again, we thank the reviewer for their constructive feedback and for recognizing the contributions and novelty of our work.

---

### Author Response · Authors · 2024-11-27
**Revised Manuscript Submission**

We sincerely thank you for your insightful feedback and constructive suggestions, which have significantly contributed to improving our manuscript titled "Black-Box Adversarial Attack on Dialogue Generation via Multi-Objective Optimization" (ID: 2921). Below, we summarize the key revisions made to address the reviewers' comments and ensure alignment with the conference's standards:

### **1. Addition of Equation Numbers**
To improve readability and facilitate reference to key mathematical components, we have added equation numbers to all equations in the manuscript. This allows for easier navigation and discussion of specific equations, especially when referring to methodology and technical details.

### **2. Evaluation of Computational Costs**
A new section, **Overview of Computational Costs (Appendix D)**, has been added to compare the runtime and query requirements of DGSlow and DGAttack under consistent conditions. We detail the computational costs incurred by each method, including:
- The total runtime for an entire dataset.
- Runtime per sample.
- The number of queries required per adversarial input.

Additionally, experiments in **Appendix D** demonstrate that reducing the population size in DGAttack (from 20 to 13–15 candidates) significantly reduces runtime (to approximately **16–18 hours**) and query requirements (to **65–75 per sample**) while maintaining high attack effectiveness. These analyses provide a clearer understanding of the trade-offs between the methods and highlight DGAttack's potential for cost-efficiency with careful parameter tuning, despite its higher computational costs due to its black-box nature.

### **3. Standard Deviations of Reported Metrics**
In **Appendix E**, we report standard deviations for all metrics (**GL, BLEU, ROUGE, METEOR, ASR, Cos**) presented in Table 1, 3 and 4. This addition assesses the stability of results across random seeds and datasets, addressing concerns about variability and robustness.

### **4. Close-Source Model Experiments**
**Appendix F** presents experiments on GPT-4o-mini, a close-source model, comparing DGAttack's transferability with DGSlow. The results demonstrate DGAttack’s superior performance in generating effective adversarial samples without model-specific knowledge, underscoring its practicality for real-world scenarios.

### **5. Ethics Statement**
The ethics statement in **Appendix G** has been expanded to address concerns regarding potential misuse of DGAttack. We emphasize:
- The untargeted nature of DGAttack, which focuses on disrupting response coherence rather than generating harmful content.
- The importance of studying vulnerabilities to motivate research on robust defenses.
- The alignment of our work with ethical standards in adversarial machine learning.

### **6. Adversarial Defense and Mitigation Strategies**
**Appendix H** outlines several potential defense mechanisms against DGAttack and similar methods, including:
- **Adversarial Training:** To improve model robustness.
- **Input Validation and Denoising:** To mitigate adversarial perturbations.
- **Robust Optimization:** Techniques to enhance resilience.
- **Detection Pipelines:** To flag anomalous behavior indicative of attacks.

These strategies are intended to stimulate further research on defense paradigms and ensure the safe deployment of DG models.

---

### Meta-Review · Area_Chair_T4sf · 2024-12-20

**Metareview:**

This paper proposes a blackbox approach to attacking language models. The proposed method relies on two objectives: (1) generation length, and (2) response coherence. Experimental results show that the attack is successful compared to a variety of baselines on triggering the models. Unfortunately, the paper is decoupled from the recent literature in a way that is grounds for rejection. First of all, the related work (and the experimental evaluations) does not capture any of the recent baselines for adversarial attacks, such as FLIRT (mehrabi et al., 2024) or PAIR (Chao et al., 2023). The language models considered are also not state of the art, and the metrics and benchmarks also don't capture the developments of the past few years, and overall the paper reads as if it was written four years ago. Even though the reviewers and the AC found that the idea of the paper is interesting, the mentioned shortcomings require major revisions that need to be addressed for the next iteration of the paper and unfortunately the paper needs to be recommended to be rejected in its current form.

Ninareh Mehrabi, Palash Goyal, Christophe Dupuy, Qian Hu, Shalini Ghosh, Richard Zemel, Kai-Wei Chang, Aram Galstyan, and Rahul Gupta. 2024. FLIRT: Feedback Loop In-context Red Teaming. In Proceedings of the 2024 Conference on Empirical Methods in Natural Language Processing, pages 703–718, Miami, Florida, USA. Association for Computational Linguistics.

Chao, P., Robey, A., Dobriban, E., Hassani, H., Pappas, G.J. and Wong, E., 2023. Jailbreaking black box large language models in twenty queries. arXiv preprint arXiv:2310.08419.

**Additional Comments On Reviewer Discussion:**

There were multiple issues raised by the reviewers that got discussed with authors. While the novelty issues were resolved in internal discussions, the paper still suffers from major issues at this point.

---

### Decision · Program_Chairs · 2025-01-22

Reject